# Beyond Scores: Proximal Diffusion Models

**Zhenghan Fang**
Mathematical Institute for Data Science
Johns Hopkins University
`zfang23@jhu.edu`

**Mateo Díaz**
Mathematical Institute for Data Science
Johns Hopkins University
`mateodd@jhu.edu`

**Sam Buchanan**
Toyota Technological Institute at Chicago
`sdbuchanan@berkeley.edu`

**Jeremias Sulam**
Mathematical Institute for Data Science
Johns Hopkins University
`jsulam1@jhu.edu`

## Abstract

Diffusion models have quickly become some of the most popular and powerful generative models for high-dimensional data. The key insight that enabled their development was the realization that access to the score—the gradient of the log-density at different noise levels —allows for sampling from data distributions by solving a reverse-time stochastic differential equation (SDE) via forward discretization, and that popular denoisers allow for unbiased estimators of this score. In this paper, we demonstrate that an alternative, backward discretization of these SDEs, using proximal maps in place of the score, leads to theoretical and practical benefits. We leverage recent results in *proximal matching* to learn proximal operators of the log-density and, with them, develop Proximal Diffusion Models (`ProxDM`). Theoretically, we prove that $\widetilde{\mathcal{O}}(d/\sqrt{\varepsilon})$ steps suffice for the resulting discretization to generate an $\varepsilon$-accurate distribution w.r.t. the KL divergence. Empirically, we show that two variants of `ProxDM` achieve significantly faster convergence within just a few sampling steps compared to conventional score-matching methods.

## 1 Introduction

Within a remarkably brief period, diffusion models [72, 27] have overtaken generative artificial intelligence, emerging as the *de facto* solution across diverse domains including medical imaging [13], video [36] and audio [28], drug discovery [15], and protein docking prediction [12]. Diffusion models combine a forward diffusion process that converts the data distribution we wish to sample into noise and a reverse-time process that reconstructs data from noise. The forward process is easy to simulate by progressively adding noise to data. The reverse process, which allows for sampling data from noise, involves the so-called score function—the gradient of the log-density of the noisy data at any time point. Though unknown in general, such a score can be estimated by common denoising algorithms that approximate minimum mean-squared-error solutions. Various diffusion algorithms have been recently proposed [75, 53, 81], all relying on forward discretizations of the reverse stochastic differential equation, and allowing for impressive generative modeling performance.

Despite these capabilities, current state-of-the-art diffusion-based methods face drawbacks: high-quality samples require a large number of model evaluations, which tends to be computationally expensive, and they are sensitive to hyperparameter choices and the lack of regularity of the data distribution [67, 35]. Naturally, computational cost, and sensitivity to hyperparameters and data regularity are not unique to diffusion models. Many numerical analysis and optimization algorithms grounded on forward discretizations of continuous-time processes share similar limitations. For

39th Conference on Neural Information Processing Systems (NeurIPS 2025).

instance, when optimizing a smooth convex function $f$, the go-to algorithmic solution, gradient descent (GD), which updates $x_+ \leftarrow x - h\nabla f(x)$, represents a forward Euler discretization of gradient flow $\dot{x} = -\nabla f(x)$ [1]. GD is known to converge when the learning rate $h$ is on the order of $1/L$ where $L$ is the gradient Lipschitz constant. Consequently, when $f$ approaches nonsmoothness (i.e., $L$ is large), GD must proceed with very small steps, thus converging slowly.

Backward discretization schemes for continuous-time processes offer a potential remedy to these issues. For example, the backward Euler discretization for gradient flow updates iterates via the implicit equation $x_+ = x - h\nabla f(x_+)$. This algorithm is known as the Proximal Point Method, and, remarkably, it converges for any positive $h$ [62]. The tradeoff is that $x_+$ is defined implicitly, requiring solving an auxiliary sub-problem that is potentially as expensive as the original one. However, in the context of generative models, where $f$ represents the log-density of data distribution, one could learn such an update directly from data via neural networks (instead of learning the score), making the per-step complexity of a backward discretization comparable to that of a forward discretization. This motivates the central question of this work.

> Can backward discretization improve diffusion models?

In this paper, we give a partial affirmative answer to this question. We develop a diffusion sampler based on an implicit, backward discretization scheme of the reverse-time stochastic differential equation (SDE). Our approach leverages recent results that allow us to train proximal operators from data via *proximal matching* [23]. We study two variants: one relying exclusively on backward discretization, and another hybrid approach that combines backward and forward terms. Our methods are competitive with score-based diffusion models—including those based on probability flow—when using a large number of discretization steps, while significantly outperforming them with fewer steps. Additionally, we provide supporting convergence theory that yields state-of-the-art rates under the idealized assumption of having access to the exact implicit update or proximal operator. Employing backward discretization of stochastic processes has been studied in both earlier [60, 5, 20, 79, 66, 42, 68, 29] as well as more recent [11, 48, 22, 49, 85, 26, 39, 50, 58, 80] works, mostly in the context of proximal version of Langevin dynamics. To our knowledge, no backward discretization has yet been explored for diffusion models. We defer a more thorough discussion of related works to after the introduction of the necessary background. Code for reproducing the experiments in this work is available at https://github.com/ZhenghanFang/ProxDM.

## 2 Background

**Diffusion and score-based generative models.** Diffusion models perform generative modeling by posing a reversed stochastic differential equation (SDE) of a process that transforms a sample from the data distribution, $X_0 \sim p_0$, $X_0 \in \mathbb{R}^d$, into Gaussian noise [72, 27]. A widely adopted SDE for the forward process is given by

$$dX_t = -\tfrac{1}{2}\beta(t)X_t dt + \sqrt{\beta(t)}dW_t, \quad t \in [0, T]. \tag{1}$$

where $W_t$ is a standard Wiener process in $\mathbb{R}^d$ (i.e. standard Brownian motion) [75]. At inference time, one can produce a new sample from $p_0$ by reversing the SDE above, resulting in the stochastic process

$$dX_t = \left[-\tfrac{1}{2}\beta(t)X_t - \beta(t)\nabla \ln p_t(X_t)\right] dt + \sqrt{\beta(t)}d\bar{W}_t, \tag{2}$$

where $\bar{W}_t$ is a reverse-time Brownian motion (with time flowing backwards from $T$ to $0$). This process can be simulated as long as the score function, $s_t(x) = \nabla \ln p_t(x)$, can be computed (or approximated for unknown real-world distributions).

The continuous-time process in (2) must be discretized to be implemented. By far, the most common strategy is applying the forward Euler-Maruyama discretization, leading to the score-based sampling algorithm:

$$X_{k-1} = X_k + \gamma_k \left[\tfrac{1}{2}X_k + \nabla \ln p_{t_k}(X_k)\right] + \sqrt{\gamma_k}z_k, \tag{Euler-Maruyama}$$

where $\gamma_k = \beta(t_k)(t_k - t_{k-1})$ and $z_k \sim \mathcal{N}(0, I)$. Many variations of these processes have been proposed. For instance, instead of the variance-preserving formulation of the process in (1), one

could instead consider a variance-exploding alternative (by removing the drift) [72, 75]. Likewise, instead of considering stochastic processes, one can find deterministic ordinary differential equations (ODEs) that reverse the forward process and match the marginals of the reversed SDE, resulting in a probability-flow ODE [75, 70].

In order to apply these sampling algorithms for generative modeling on real-world data, corresponding score functions must be obtained for every distribution $p_t$. The popular *score matching* approach [32, 72, 74, 59] learns parametric regression functions $s_\theta(x, t)$ (with parameters $\theta$) by minimizing the objective

$$\mathbb{E}_{t, X_0, \varepsilon} \left\{ \lambda(t) \left\| s_\theta(X_t, t) - \frac{-\varepsilon}{\sqrt{1 - \alpha_t}} \right\|_2^2 \right\}, \tag{3}$$

where $X_0 \sim p_0$, $\varepsilon \sim \mathcal{N}(0, I_d)$, $\alpha_t = \exp\left(-\int_0^t \beta(s)ds\right)$, $X_t = \sqrt{\alpha_t}X_0 + \sqrt{1 - \alpha_t}\varepsilon$, and $\lambda(t)$ is a function that weighs different terms (for different $t$) appropriately. It can be shown [75] that the solution to such an optimization problem results in the desired score, $s^\star(x, t) = \nabla \ln p_t(x)$. There are two important remarks about this approach: $(i)$ such a function amounts to a *denoiser*—that is, it seeks to recover a signal corrupted by (Gaussian) noise[1]—and $(ii)$ such a denoiser is a minimum mean-squared error (MMSE) estimate, as it minimizes the reconstruction error with respect to a squared Euclidean norm. This can also be seen alternatively by Tweedie's formula [21, 61], which relates the score to the conditional expectation of the unknown, clean sample $\sqrt{\alpha_t}X_0$; namely $\mathbb{E}[\sqrt{\alpha_t}X_0 \mid X_t] = X_t + \sigma^2 \nabla \ln p_t(X_t)$, where $\sigma = \sqrt{1 - \alpha_t}$. Such a conditional mean is the minimizer of the MSE regression problem [57], showing that MMSE denoisers provide direct access to the score of the distribution $p_t$. Fortunately, the design and development of denoising algorithms is very mature, and strong algorithms exist [57]. As an example, it is standard to employ convolutional networks with U-net architectures [65] for score-matching in image generation [27, 75].

**Proximal operators.** Proximal operators are ubiquitous in optimization, signal and image processing, and machine learning. For a functional $f \colon \mathbb{R}^d \to \mathbb{R}$, the proximal operator of $f$ is a mapping from $\mathbb{R}^d$ to $\mathbb{R}^d$ defined by

$$\text{prox}_f(x) = \arg\min_u f(u) + \tfrac{1}{2}\|u - x\|_2^2. \tag{4}$$

Intuitively, the proximal operator finds a point that is close to the anchor, $x$, and has low function value in $f$. It is convenient to introduce an explicit parameter $\lambda \in \mathbb{R}^+$ to control the relative strength of the functional $f$, and consider $\text{prox}_{\lambda f}(x)$ instead. In this way, $\text{prox}_{\lambda f}(x) \to x$ when $\lambda \to 0$, whereas $\text{prox}_{\lambda f}(x) \to \arg\min_u f(u)$ when $\lambda \to \infty$. Further, when $f$ is $L$-smooth and $\lambda \leq \frac{1}{L}$, first-order optimality conditions yield that $x^+ = \text{prox}_{\lambda f}(x)$ if, and only if, $x^+ = x - \lambda \nabla f(x^+)$, which illustrates its relation to backward discretizations.

Importantly, proximal operators can be equivalently described as *maximum-a-posteriori* (MAP) denoising algorithms under a Gaussian corruption model. Indeed, it is easy to verify[2] that when $y = x + \varepsilon$, for $x \sim p_0$, Gaussian noise $\varepsilon \sim \mathcal{N}(0, \sigma^2 I_d)$, and $f = -\ln p_0$, then $\text{prox}_{\sigma^2 f}(y)$ provides the mode of the conditional distribution of the unknown $x$ given the measurements $y$; i.e.,

$$\text{prox}_{\sigma^2 f}(y) = \arg\max_X p_0(X|y). \tag{5}$$

This view of proximal maps in terms of MAP denoising algorithms should be contrasted with that of the score, which relies on MMSE denoising algorithms: unlike the latter, the former does not require the function $f = -\ln p_0$ to be differentiable nor continuous.[3] This has been extensively exploited in nonsmooth optimization [3, 14, 18, 19], leading to speedups in convergence. The advantages of employing proximal operators (instead of gradients) for sampling have also been recognized in earlier works [60, 5, 20, 79, 26, 39], leading to faster convergence or less stringent conditions on the parameters of the problem, like step sizes. More generally, MAP estimates are known to provide samples that are closer to the support of the data distribution than MMSE denoisers [16]. In the following section, we will harness proximal maps to develop a new discretization of denoising diffusion models, leveraging MAP denoisers instead of MMSE ones.

---

[1]More precisely, (3) seeks to equivalently estimate the *noise* that has been added to the ground truth $X_t$.

[2](5) follows since $p_0(X|y) \propto p(y|X)p_0(X) \propto \exp(-\|y-X\|^2/(2\sigma^2)) \exp(-f(X))$ and ln is monotonic.

[3]It simply requires $f$ to be weakly-convex or prox-regular [63].

---

**Algorithm 1** Proximal Diffusion Model (`ProxDM`)

---

**Input:** Map $\text{prox}_{-\lambda \ln p_t}(\cdot)$, function $\beta(\cdot)$, grid $\{0 = t_0 < t_1 < \cdots < t_N = T\}$, boolean `hybrid`.
**Output:** Sample $X_0$.

    Generate $X_N \sim \mathcal{N}(0, I)$
    **for** $k = N$ **to** 1
        Generate $z_k \sim \mathcal{N}(0, I)$ and compute $\gamma_k \leftarrow \int_{t_{k-1}}^{t_k} \beta(s) ds$
        **if** `hybrid` **then**
            Update $X_{k-1} \leftarrow \text{prox}_{-\gamma_k \ln p_{t_{k-1}}} \left[ \left(1 + \frac{1}{2}\gamma_k\right) X_k + \sqrt{\gamma_k} z_k \right]$
        **else**
            Update $X_{k-1} \leftarrow \text{prox}_{-\frac{2\gamma_k}{2-\gamma_k} \ln p_{t_{k-1}}} \left[ \frac{2}{2-\gamma_k} \left( X_k + \sqrt{\gamma_k} z_k \right) \right]$
    **return** $X_0$

---

## 3 Proximal-based generative diffusion modeling

Recall from Section 2 that all score-based generative models apply forward discretization to the reverse-time SDE, resulting in updates of the general form in (Euler-Maruyama), with the score playing a central role. We now propose a different avenue to turn this SDE into a sampling algorithm.

### 3.1 From backward discretization to proximal-based sampling

Consider now the *backward* discretization of the reversed-time SDE from (2), that is:

$$X_{k-1} = X_k + \gamma_k \left[ \tfrac{1}{2} X_{k-1} + \nabla \ln p_{t_{k-1}} (X_{k-1}) \right] + \sqrt{\gamma_k} z_k. \tag{6}$$

Unlike the forward (Euler-Maruyama) discretization, this is an implicit equation of $X_{k-1}$. It is easy to verify that this update corresponds to the first-order optimality conditions of the problem $\min_u g(u)$, where $g(u) := -\frac{2\gamma_k}{2-\gamma_k} \ln p_{t_{k-1}}(u) + \frac{1}{2} \left\| u - \frac{2}{2-\gamma_k}(X_k + \sqrt{\gamma_k} z_k) \right\|_2^2$, precisely a proximal step. Such an update can be written succinctly as

$$X_{k-1} = \text{prox}_{-\frac{2\gamma_k}{2-\gamma_k} \ln p_{t_{k-1}}} \left[ \frac{2}{2 - \gamma_k} (X_k + \sqrt{\gamma_k} z_k) \right]. \tag{PDA}$$

We refer to this step as the proximal diffusion algorithm (PDA). It remains to define the discretization of the step $\gamma_k$: recall that for forward discretizations, one can take $\gamma_k = \beta(t_k) \Delta t_k$. Here, we instead make the choice of setting $\gamma_k = \int_{t_{k-1}}^{t_k} \beta(s) ds$. Note that both are equivalent when $\Delta t_k \to 0$.

It is worth remarking that the discretization in (6) is a *fully* backward discretization: one where both terms depending on the time-continuous $X(t)$ are discretized as $X_{k-1}$. Yet, one can also consider a *hybrid* approach by considering the updates given by $X_{k-1} = X_k + \gamma_k \left[ \frac{1}{2} X_k + \nabla \ln p_{t_{k-1}} (X_{k-1}) \right] + \sqrt{\gamma_k} z_k$, where both $X_k$ and $X_{k-1}$ are employed in the discretization of the drift term. In this case, the resulting update can be concisely written as

$$X_{k-1} = \text{prox}_{-\gamma_k \ln p_{t_{k-1}}} \left[ \left(1 + \tfrac{1}{2} \gamma_k\right) X_k + \sqrt{\gamma_k} z_k \right]. \tag{PDA-hybrid}$$

The resulting hybrid approach is akin to a forward-backward method in nonsmooth optimization [3], is complementary to the update in (PDA), and will have advantages and disadvantages, on which we will expand shortly. For now, we have everything we need to formally define our sampling algorithm for Proximal Diffusion Models (`ProxDM`), as presented in Algorithm 1.

**Remarks.** First, we note that this technique for deriving proximal versions of diffusion models via backward discretization is quite general and widely applicable to other variants of diffusion sampling schemes. We have shown this idea for a variance preserving (VP) SDE and the Euler-Maruyama solver, but the same can be done for the reverse diffusion sampler and the ancestral sampler (see [75, Appx. E]), as well as variance exploding (VE) SDE and probability flow ODE [75]. In each case, one would obtain a new proximal-based sampling algorithm that mirrors the diversity of score-based sampling algorithms (see further discussions in Appendix F).

Compare now the update in (PDA) to the score (gradient)-based update in (Euler-Maruyama). First, both algorithms call a denoiser of the data distribution at each step; the score-based method uses an MMSE denoiser based on gradients, whereas PDA uses a MAP denoiser via proximal operators. Second, both updates consist of a denoising step and a noise addition step, but in switched order: the forward version applies denoising first, followed by noise addition. As a result, the final samples from score-based models do contain small amounts of noise, requiring an additional, ad-hoc denoising step at the end to mitigate this and improve sampling performance (see e.g., [75, Appx. G]). In contrast, PDA performs denoising after noise injection. This avoids the need for extra modifications and yields a more principled and general scheme.

Given the benefits of backward discretization, why should one bother with a hybrid approach (PDA-hybrid)? As we will see in the next section, (PDA) enjoys faster convergence rates. However, this comes at the cost of a limitation in the minimal number of sampling steps: one can verify from (PDA) that we must ensure $\gamma_k < 2$ (as otherwise the regularization parameter, $\frac{2\gamma_k}{2-\gamma_k}$, would be negative), resulting in an upper-bound on the step size and thus a lower-bound to the number of sampling steps. The hybrid approach in (PDA-hybrid) does not suffer from this hard constraint, resulting in practical benefits, albeit at the expense of a slower theoretical rate.

The reader should note that the connection between backward discretization and proximal-based algorithms is not new and has been explored in different contexts across optimization and sampling. For example, while forward discretization of the gradient flow leads to the gradient descent algorithm, backward discretization corresponds to the proximal point method [62]. In the context of sampling, forward discretization of the Langevin dynamics yields the (unadjusted) Langevin algorithm (ULA), whereas backward discretization leads to the proximal Langevin algorithm (PLA) [60, 5, 20, 79]. It is interesting to note that, in this context, proximal algorithms that leverage backward discretizations often provide benefits that the simple forward ones do not have: they are often more stable to the choice of hyper-parameters and can do away with requiring differentiability of the potential function [60, 79]. A less related but interesting connection between forward-backward methods (in the space of measures) and sampling has also been studied in recent works [77, 42, 68, 11, 80], and can provide unbiased asymptotic samples (which ULA does not provide) [80]. These advances, however, have all been provided in the context of Langevin dynamics and, to the best of our knowledge, our PDA approach in Algorithm 1 is the first instantiation of these ideas in the context of diffusion processes.

Lastly, score-based methods need the gradient of the log-density at time $t_k$, $\nabla \ln p_{t_k}$, which can be easily estimated via score-matching. On the other hand, both proximal updates in Algorithm 1 require $\text{prox}_{-\lambda_k \ln p_{t_{k-1}}}$, i.e. the proximal operator for the log-density at time $t_{k-1}$. Next, we show how to estimate such proximal operators.

### 3.2 Proximal matching for proximal operator estimation

Learning data-dependent proximal operators has only recently begun to receive attention [30, 56, 47, 23]. We will make use of recent work by Fang et al. [23], who presented a general approach to obtaining learned proximal networks (LPNs). They present a *proximal-matching* loss function that ensures the recovery of the proximal of the log-density[4] of continuous distributions at arbitrary regularization strengths from i.i.d. samples from that distribution. In a nutshell, proximal-matching augments the random samples $X \sim p_0$ with Gaussian noise, $Y = X + \sigma\epsilon$, for $\epsilon \sim \mathcal{N}(0, I_d)$. Then, it performs a regression using a special loss function and schedule to learn a MAP denoiser for the data distribution, with the noise level dependent on the desired regularization strength $\lambda$ in the proximal. To learn $\text{prox}_{-\lambda \ln p_0}$, one minimizes the problem $\mathbb{E}_X \mathbb{E}_{Y|X} [\ell_{\text{PM}}(f_\theta(Y), X; \zeta)]$ with the proximal-matching loss

$$\ell_{\text{PM}}(x, y; \zeta) = 1 - \exp\left(-\frac{\|x - y\|_2^2}{d\zeta^2}\right).$$

Fang et al. [23, Theorem 3.2] showed that, in the limit of $\zeta \to 0$, the problem above[5] recovers the proximal operator of the log density of the distribution with regularization strength $\sigma^2$; i.e.

---

[4]The work in [23] also proposes a specific parametrization of neural networks based on gradients of convex functions (such as those arising from ICNNs [2]) that guarantees exact proximal operators for any obtained network.

[5]We remark that the form of $\ell_{\text{PM}}$ used here differs slightly from the definition in [23], but it does not change the optimization problem nor its guarantees.

$f_{\theta^*} = \mathrm{prox}_{-\sigma^2 \ln p_0}$. Intuitively, the proximal matching loss can be interpreted as a smoothed version of the $\ell_0$ pseudo-norm. In one dimension, minimizing the (non-differentiable) $\ell_0$ loss recovers the mode of a distribution. The proximal matching loss generalizes this idea to higher dimensions and differentiable functions, making it amenable to first-order optimization methods. Fig. 6 visualizes the proximal matching loss along with the mean-squared error and $\ell_1$ losses to illustrate their difference.

Here, we extend proximal-matching to a setting that will enable us to learn a set of proximals for varying densities and regularization strengths, as needed for ProxDM: the set $\{\mathrm{prox}_{-\lambda \ln p_t}\}$ for a range of $t$ and $\lambda$'s. Our generalization of the proximal matching objective for proximal diffusion model is

$$\theta^* = \arg\min_\theta \mathbb{E}_{t,\lambda} \left\{ \mathbb{E}_{X_t} \mathbb{E}_{Y|X_t} \left[ \ell_{PM} \left( f_\theta(Y; t, \lambda), X_t; \zeta \right) \right] \right\} \tag{7}$$

where $X_t \sim p_t$ and $Y|X_t \sim \mathcal{N}(X_t, \lambda I_d)$ are clean and noisy samples for the forward process, respectively. Intuitively, with sufficient data and model capacity (just as in score-matching networks), the optimal solution to (7), denoted by $f_{\theta^*}(x; t, \lambda)$, approximates $\mathrm{prox}_{-\lambda \ln p_t}(x)$ as $\zeta \to 0$.

### 3.3 Practical considerations and implementation details

Although this proximal matching framework is conceptually simple, translating it into practice requires careful implementation. Herein, we present a few important methodological details.

**Sampling of** $(t, \lambda)$ **pair.** Optimizing (7) requires sampling $(t, \lambda)$ pairs, and the sampling distribution should ideally match the distribution to be used after training. As in score-based models, a reasonable choice for the time step $t$ is uniform in $[0, T]$. The values of $\lambda$ is less obvious, however. Recall that this regularization strength depends on the step size ($\gamma_k$) and, thus, also on time $t_{k-1}$ and $t_k$. We adopt a heuristic for sampling $(t, \lambda)$ according to the candidate number of steps to be used at inference time (see Appendix E.2 for further details and motivation), which in practice enables the trained model to be applied for a range of different numbers of steps. We note that this sampling scheme is not universally optimal, and we leave the optimization of sampling schemes to future work.

**Parameterization of the proximal network** $f_\theta(x; t, \lambda)$**.** We parameterize the proximal operator $\mathrm{prox}_{-\lambda \ln p_t}$ using a neural network denoted by $f_\theta(\cdot; t, \lambda)$. Notably, while the score network is conditioned on a single time scalar $t$ [27, 75], the proximal network is conditioned on two scalars, $t$ and $\lambda$. Similar to score networks, conditioning is implemented by adding learned embeddings for $t$ and $\lambda$ to the intermediate feature maps. Moreover, the score network indirectly parameterizes the score via $s_\theta(x, t) = \frac{-\epsilon_\theta(x,t)}{\sqrt{1-\alpha_t}}$ [27], where $s_\theta(x, t)$ is the (learned) score and $\epsilon_\theta(x, t)$ is a neural network. Analogously, we parameterize the proximal indirectly by letting $f_\theta(x; t, \lambda) = x - \sqrt{\lambda}\epsilon_\theta(x; t, \lambda)$, where $\epsilon_\theta(x; t, \lambda)$ is a neural network that predicts the residual of the proximal operator: $\epsilon_\theta(x; t, \lambda) = [x - f_\theta(x; t, \lambda)]/\sqrt{\lambda}$. This parameterization led to better empirical results at large step numbers in our experiments and coincides naturally with the balancing of objective across different choices of $(t, \lambda)$'s, which we discuss next.

**Balancing contributions of different** $(t, \lambda)$ **in the objective.** Score matching uses a weighing function to balance the loss at different times (see [75, $\lambda(t)$ in Eq. (7)]). For proximal matching, a proper weighing should also be used for balanced learning, and the weighing should be based on both $t$ and $\lambda$ (c.f. (7)). A general rule of thumb would choose these weights such that the objective of the inner expectation in (7) has similar magnitude across different $t$ and $\lambda$. Score model achieves this by weighing according to the square of the magnitude of the target in the objective [75, Sec. 3.3], which is reasonable when the magnitude of the target can be estimated based on $t$ and when the loss function is (2-)homogeneous. Unfortunately, none of these are true for proximal-matching. Thus, instead of weighing the objective, we transform the target to align its magnitude across different $(t, \lambda)$'s. Specifically, we rewrite the problem in (7) as follows (but without changing the objective):

$$\theta^* = \arg\min_\theta \mathbb{E}_{t,\lambda} \left\{ \mathbb{E}_{X_t} \mathbb{E}_{Y|X_t} \left[ \ell_{PM} \left( \frac{Y - f_\theta(Y; t, \lambda)}{\sqrt{\lambda}}, \frac{Y - X_t}{\sqrt{\lambda}}; \zeta \right) \right] \right\}, \tag{8}$$

where $\epsilon := \frac{Y - X_t}{\sqrt{\lambda}}$ is the new target. Since $Y \sim \mathcal{N}(X_t, \lambda I)$, we have $\epsilon \sim \mathcal{N}(0, I)$. Notably, the magnitude of $\epsilon$ is *independent* of $(t, \lambda)$, allowing for a natural balance. Plugging in the parameterization

of $f_\theta(x; t, \lambda) = x - \sqrt{\lambda}\epsilon_\theta(x; t, \lambda)$, the final objective reduces to:

$$\theta^* = \arg\min_\theta \mathbb{E}_{t,\lambda} \left\{ \mathbb{E}_{X_t} \mathbb{E}_\epsilon \left[ \ell_{\mathrm{PM}} \left( \epsilon_\theta \left( X_t + \sqrt{\lambda}\epsilon \, ; t, \lambda \right), \epsilon \, ; \zeta \right) \right] \right\}. \tag{9}$$

Recall that $\epsilon_\theta$ is a neural network, $f_\theta(x; t, \lambda)$ is the learned approximation of $\mathrm{prox}_{-\lambda \ln p_t}(x)$, $(t, \lambda)$ are sampled according to a predefined distribution, and $\zeta$ is a hyperparameter that should decrease through training. Note that this objective is similar to that of diffusion models with analogous parameterization [27, Eq. (12)]. We include pseudo-code for proximal-matching training in Algorithm 2, noting that it simply boils down to optimizing (9) in a stochastic manner, allowing for any solvers of choice (stochastic gradient descent, Adam, etc).

It is worth comparing the resulting method with score matching. First, score matching samples Gaussian noise only once (to compute $X_t$). In contrast, our approach samples noise *twice*: once to construct $X_t$ and once for proximal matching training. Second, score matching aims to learn the MMSE denoiser for $X_0$ (more precisely, $\sqrt{\alpha_t}X_0$) with noise level $\sqrt{1 - \alpha_t}$, whereas proximal matching searches for the MAP denoiser for $X_t$ with noise level $\sqrt{\lambda}$. Lastly, the loss in proximal matching is parameterized by $\zeta$, which decreases via a scheduler, whereas score matching uses a fixed MSE loss throughout.

## 4 Convergence theory

In this section, we establish a convergence guarantee for Algorithm 1 under the idealized assumption that the proximal operator we use is exact instead of learned. Our results provide a bound on the number of steps needed by the two discretizations we proposed—backward and hybrid—to achieve a KL-divergence of less than $\varepsilon$. We consider the canonical Ornstein–Uhlenbeck (OU) process as the forward process, that is $\beta(t) \equiv 2$ in (1), as is standard in both practical implementations of diffusion models and theoretical analyses of convergence. Further, we consider a uniform time grid $t_k = kT/N$. With this choice, letting $h := T/N$, the step sizes $\gamma_k$ in Algorithm 1 reduce to $\gamma_k = 2h$.

We now state an informal version of our main result. The formal statement requires additional technical details that we defer to Appendix C. Recall that $p_0$ is the target distribution.

**Theorem 1** (Informal). *Suppose that $\mathbb{E}_{p_0}\|X\|^2$ is bounded above by $M_2 < \infty$. Further, assume for all $t \geq 0$, the potential $\ln p_t$ is a three-times differentiable function with $L$-Lipschitz gradient and $H$-Lipschitz Hessian, the moment $\mathbb{E}_{p_t}\|\nabla \ln p_t(X)\|^2$ is bounded above by $C_p < \infty$, and certain regularity conditions hold. Set the step size and time horizon to satisfy $h \leq \frac{1}{8L+4}$ and $T \geq 0.25$. Let $q_T$ be the distribution of the output of Algorithm 1. The following two hold true.*

*(**Hybrid**) If the* `hybrid` *flag is set to true, then*

$$\mathrm{KL}(p_0\|q_T) \lesssim (d + M_2)e^{-T} + hTdL^2 + h^2T\left[(d + M_2 + C_p)L^2 + d^2H^2\right] + h^4Td^3L^2H^2.$$

*(**Backward**) If the* `hybrid` *flag is set to false and further $h \leq 1$, then*

$$\mathrm{KL}(p_0\|q_T) \lesssim (d + M_2)e^{-T} + h^2T\left[(d + M_2 + C_p)L^2 + d^2H^2\right] + h^4Td^3L^2H^2.$$

A few remarks are in order. First, the uniform bounds on $H$ and $C_p$ can be relaxed to a much weaker bound on an appropriate running average on $t$—we elaborate on the more general assumptions in the formal statement of the result. Second, the regularity conditions mentioned above—made precise in the appendix—are necessary to guarantee that certain boundary conditions are met. Interestingly, similar conditions seem to be implicitly assumed in many existing results [6, 79, 41, 78], and we defer a longer discussion to the appendix. Third, note that we do not require the distribution to be log-concave, nor do we impose that it satisfies the log-Sobolev inequality. Finally, the proof strategy is inspired by the analysis of the Proximal Langevin algorithm from [79], in combination with techniques from [6, 7, 40, 41, 78]. To better illustrate this result, the following corollary bounds the number of steps necessary to achieve an $\varepsilon$-accurate distribution in KL divergence.

**Corollary 1.** *Consider the setting of Theorem 1. Moreover, assume that $M_2 \lesssim d, C_p \lesssim dL^2$. The following two hold true.*

*(**Hybrid**) If the* `hybrid` *flag is set to true, then $\mathrm{KL}(p_0\|q_T) \leq \varepsilon$ provided that*

$$T \gtrsim \log\left(\frac{d}{\varepsilon}\right) \quad and \quad N \gtrsim \frac{T^2dL^2}{\varepsilon} + \left[\frac{T^3(dL^4 + d^2H^2)}{\varepsilon}\right]^{1/2} + \left(\frac{T^5d^3L^2H^2}{\varepsilon}\right)^{1/4}.$$

***(Backward)*** *If the* `hybrid` *flag is set to false and further* $h \le 1$*, then* $\mathrm{KL}(p_0 \| q_T) \le \varepsilon$ *provided that*

$$T \gtrsim \log\left(\frac{d}{\varepsilon}\right) \quad and \quad N \gtrsim \left[\frac{T^3(dL^4 + d^2 H^2)}{\varepsilon}\right]^{1/2} + \left(\frac{T^5 d^3 L^2 H^2}{\varepsilon}\right)^{1/4}.$$

The dimension dependence on $M_2$ and $C_p$ is standard and is, e.g., satisfied by Gaussian distributions. With this result, the benefits of the full backward discretization become clear: the hybrid discretization requires $\widetilde{\mathcal{O}}(d\varepsilon^{-1})$ steps, while the backward discretization requires only $\widetilde{\mathcal{O}}(d\varepsilon^{-\frac{1}{2}})$ steps. This benefit comes at the expense of the additional condition that $h \le 1$, since otherwise the regularization coefficient of the proximal operator would become negative. This translates into a hard lower bound on the number of steps one can take, as we discussed above and will see shortly again in the experiments section. In comparison, the most competitive rates for score-based samplers in terms of convergence in KL divergence are $\widetilde{\mathcal{O}}(d\varepsilon^{-1})$ for a vanilla method [6, 4] and $\widetilde{\mathcal{O}}(d^{\frac{3}{4}}\varepsilon^{-\frac{1}{2}})$ for an accelerated strategy [81]. Note that these rates are not directly comparable, as the underlying assumptions and algorithmic setups (e.g., smoothness, boundedness, or early stopping) differ across works. In particular, [6] assumed $L$-Lipschitzness for the score function, similar to our work but without the additional smoothness assumption on the Hessian as used here. On the other hand, [4] lifted the smoothness assumptions entirely but instead analyzed the KL with respect to a slightly noise-perturbed version of the target distribution. We point interested readers to the broader body of work on the convergence theory of diffusion models, including both SDE-based samplers [7, 40, 41, 6, 4, 44, 45, 43] and ODE-based samplers [9, 8, 46]; see [44, Section 4] and [43] for a comprehensive review.

## 5  Experiments

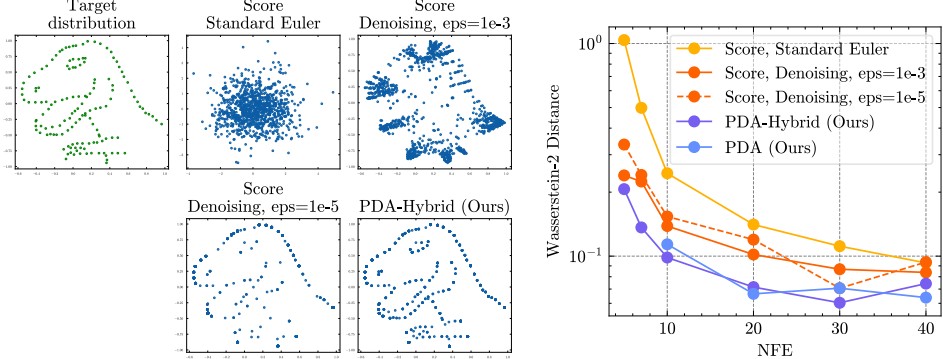

Figure 1: Sampling from a distribution supported on a discrete set of points in 2D using exact score and proximal operators. LEFT: the target distribution and samples generated by various samplers using 5 sampling steps; RIGHT: Wasserstein-2 distance between sample and target across varying step numbers (NFE, number of function evaluations). Standard Euler-Maruyama without denoising fails when using only 5 steps. Score-based sampler with a denoising step at the end, as is common [75], requires tuning $\epsilon$: the step size for the last denoising step. Both PDA variants perform well without additional hyperparameters.

**Synthetic example.**   We begin with an example in cases where the scores and proximals can be computed to arbitrary precision. We set $p_0$ as a uniform distribution over a discrete set of points (dataset from [55]), and run score-based diffusion sampling, as well as (PDA) and (PDA-hybrid) (see Algorithm 1). We include both the standard Euler-Maruyama method ("Standard Euler"), as well as the popular modification that includes an ad-hoc final denoising step after the last iterate [75]. As shown by the results in Fig. 1, both variants of `ProxDM` provide faster convergence with a closer final sample to the ground truth $p_0$, and require less hyperparameters to tune and no ad hoc modification. To further study the impact of the smoothness of the distribution on sampling performance, we perform an experiment to sample from 1D distributions with different levels of smoothness at the boundary of their support (see details and results in Appendix H.1). Although the relation between sample quality and smoothness of the distribution is relatively subtle–likely due to smoothing effect inhere in the forward diffusion process–it is clear that `ProxDM` recovers the support of the distribution more accurately than score-based samplers.

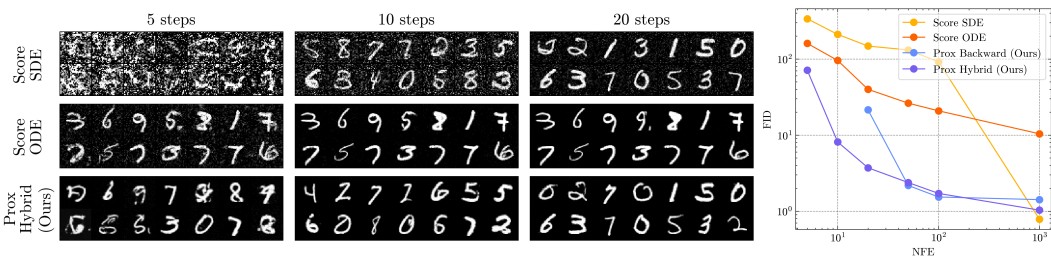

Figure 2: Left: MNIST samples generated by the score SDE sampler, score ODE sampler, and our hybrid `ProxDM`. Right: the resulting FID score as a function of the number of sampling steps.

**MNIST.** We now move to a real, simple case for digits [17], where we must train score-networks (through score matching) as well as our learned proximal networks for `ProxDM`. The $\beta(t)$ in forward process is set as linear, $\beta(t) = \beta_{\min} + (\beta_{\max} - \beta_{\min})\, t$, for $t \in [0, 1]$, with $\beta_{\min} = 0.1, \beta_{\max} = 20$, as is standard [75]. `ProxDM` is pre-trained with an $\ell_1$ loss, followed by proximal matching loss with $\zeta = 1$ in the first half and decayed to $0.5$ in the second half. Both networks are trained for the same number of epochs (see more details in Appendix E.3).

We compare with two score-based sampler: Euler-Maruyama for the reverse-time SDE (denoted as "SDE") and the Euler sampler for the probability flow ODE (resp."ODE") [75]. We show the FID score by NFE (number of function evaluations, i.e. sampling steps) and uncurated samples in Fig. 2 (right and left, respectively). `ProxDM` provides significant speedup compared to both the score samplers based on the reverse-time SDE and probability-flow ODE.

**CIFAR10.** We perform a similar experiment on the CIFAR10 dataset [37]. The score model uses the same U-Net as in [27]. The network for `ProxDM` is the same as the score network, except that it has two parallel conditioning branches for $t$ and $\lambda$ respectively, with a comparable total number of parameters. Score matching follows the standard setup in [27] with learning rate warm-up, gradient clipping, and uniform $t$ sampling as introduced in [75]. For proximal matching, the proximal network is pretrained with $\ell_1$ loss, followed by proximal matching loss with $\zeta = 2$ in the first half and $\zeta = 1$ in the second half (see more details in Appendix E.3). Fig. 3 shows the FID score vs. NFE curves. It is worth noting that the score SDE sampler's FID at 1000 steps matches the value reported in the original paper [27]. It is clear that our `ProxDM` outperforms score-based samplers and enables faster sampling. Notably, the FID of `ProxDM` at 10 steps matches that of the score ODE sampler at 20 steps and surpasses the score SDE sampler even at 100 steps.

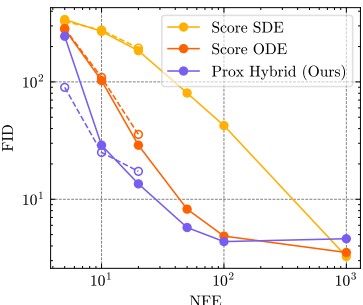

Figure 3: FID vs. number of sampling steps (NFE) on CIFAR10. The dashed lines correspond to models trained specifically for 5, 10 and 20 steps, as opposed to the full-range models in solid lines.

Uncurated samples are shown in Fig. 4, demonstrating sharper and cleaner samples by `ProxDM` with fewer sampling steps. Furthermore, when trained specifically for small step numbers (5, 10 and 20), the advantage of `ProxDM` over score-based samplers becomes more significant (see Fig. 3, dashed lines). If one is specifically interested in a very small number of steps, disregarding some of the heuristics described in Section 3.3 can lead to even better results (see Fig. 9). Additionally, we report the precision and recall metrics [38], which measure the quality of generated samples and coverage of the real data distribution, respectively (see detailed definitions in Appendix E.3). The results are presented in Fig. 10, further illustrating the improved sample quality and coverage achieved by `ProxDM`. Finally, the FID of `ProxDM` at 1000 steps appears worse than that at 100 steps–we discuss on this in Appendix A.3.

**CelebA-HQ** ($256 \times 256$). We further evaluate our approach for high-resolution image synthesis using the CelebA-HQ dataset at $256 \times 256$ resolution [34]. Because of the higher dimensionality of this data, both the score and proximal approaches operate in the latent space of a pretrained VAE, following the standard latent-diffusion setup [64]. As shown in Fig. 11, the hybrid `ProxDM` achieves improved FID compared to the score SDE sampler and performs competitively with the score ODE

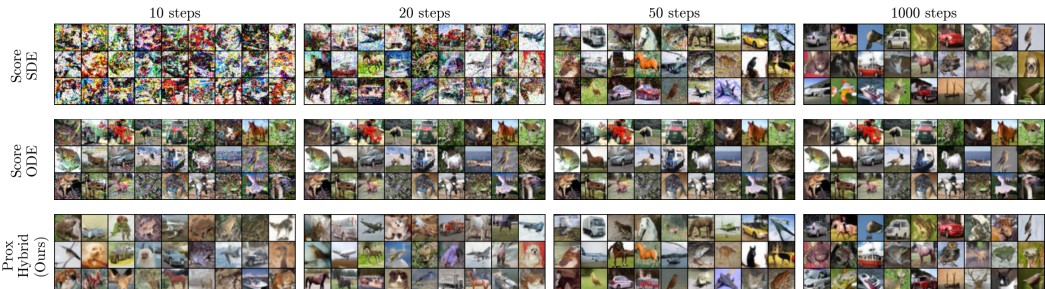

Figure 4: CIFAR10 samples from score SDE, score ODE, and hybrid `ProxDM` samplers.

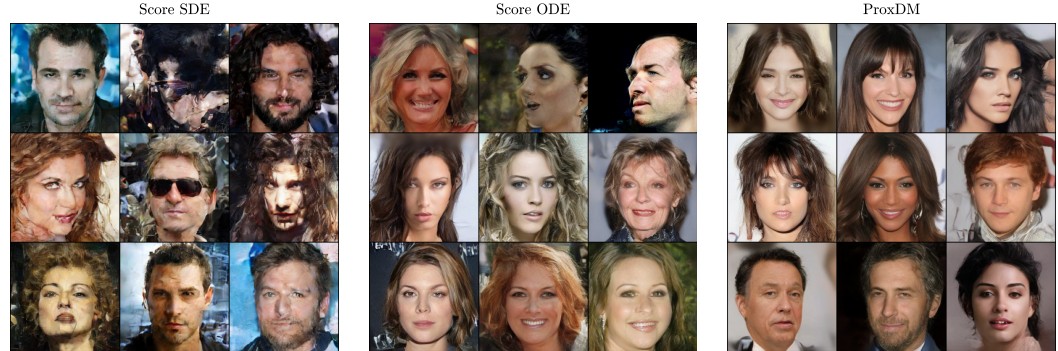

Figure 5: CelebA-HQ ($256 \times 256$) samples from score SDE, score ODE, and hybrid `ProxDM` samplers using 20 sampling steps.

sampler across all step counts. Uncurated visual samples using 20 sampling steps are provided in Fig. 5; samples at other step numbers can be found in Fig. 12.

## 6   Conclusions and limitations

Diffusion models represent a paradigm shift in generative modeling. Since a reversed-time SDE drives sampling, discretization strategies are needed to turn these into practical algorithms. To our knowledge, all diffusion models have so far relied on forward discretizations of these SDEs, relying on the distribution score at each step. Here, we have shown that backward discretizations can be greatly beneficial, both in theory and practice. For the former, our full backward `ProxDM` requires only $\widetilde{\mathcal{O}}(d/\sqrt{\varepsilon})$ steps to produce a sampling distribution $\varepsilon$-away from the target in KL divergence (assuming perfect proximal steps). In practice, we show that neural networks trained with proximal matching allow for practical methods that generate accurate samples far faster than alternative score-based algorithms, and even faster than score-based ODE samplers.

Our work has several limitations. First, our theoretical analysis assumes access to an oracle proximal operator. While we demonstrated that such models can be approximately obtained in practice, incorporating this discrepancy (say, at an $\alpha$-accuracy level) as done in other works [7, 6, 31] remains open. Second, our convergence results also require additional regularity conditions. While these are arguably mild (see discussion in the appendix), we conjecture that these could be removed with a more careful analysis. On the practical side, we provided a first instantiation of learned proximal networks for `ProxDM`, but many open questions remain, including how to choose the optimal weighing strategy for different sampling points across training, optimal ways to schedule the $\zeta$ hyper-parameter in the proximal-matching loss, and employing networks that provide exact proximal operators for any sampling time.

## Acknowledgment

ZF acknowledges support from the Johns Hopkins University Mathematical Institute for Data Science (MINDS) Fellowship. MD was partially supported by NSF awards CCF 2442615. MD and JS were partially supported by DMS 2502377. ZF and JS have been partially supported by NIH Grant P41EB031771.

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

# Contents

## A  Discussions

### A.1  Comparison to Existing Accelerated Methods of Diffusion Models

Since the introduction of diffusion models, numerous attempts have been made to developing accelerated methods, including those based on deterministic samplers [70, 8], higher-order solvers [53, 82, 45, 81], consistency models [76, 71], distillation methods [67, 10, 84, 83], flow matching [51, 52], and shortcut models [24]. Our aim was not merely to accelerate inference within score-based models, however, but to propose a new and principled framework for diffusion models based on new discretization schemes with convergence guarantees.

In particular, we demonstrate that backward discretization schemes lead to a new class of diffusion models based on proximal operators (instead of scores), and show that this formulation leads to immediate gains in sampling efficiency compared to vanilla score-based SDE samplers. Given its simplicity and generality, the same principle can be naturally extended to existing accelerated methods–for example, by applying backward discretization to probability flow ODEs and flow matching. Our approach thus represents an orthogonal direction to prior advancements, offering a general pathway to improve existing methods. Notably, our approach requires no distillation and is trained directly from data. Moreover, our method enjoys provably favorable convergence properties over score-based counterparts.

Finally, we believe the findings of `ProxDM` could lead to novel understandings of recent empirical heuristics. For example, recent works have shown the benefits of training generative models using MAP-promoting losses rather than the conventional MSE loss [71, 25]. Our framework employs the proximal matching loss, which also seeks a MAP estimate, but arises naturally from the perspective of learning proximal operators. This connection offers new theoretical insights for understanding the effectiveness of such losses and may inspire future methodological improvements.

### A.2  Connection to Maximum Likelihood Training

It has been shown that minimizing the denoising score matching objective is equivalent to maximizing a lower bound on the likelihood of the observed data [27]. In contrast, our proximal matching loss does not naturally admit such a Maximum-likelihood (MLE) interpretation, since it is derived from a continuous-time SDE perspective rather than the discrete Markov chain formulation. It remains an open question whether such an equivalence exists between proximal matching and MLE training. Notably, since our `ProxDM` differs fundamentally from score-based diffusion models in both the transition kernel (non-Gaussian vs. Gaussian) and training objective (non-MSE vs. MSE), we anticipate that establishing such an equivalence is nontrivial–if at all possible.

Despite this, the samples generated by our method enjoy strong theoretical guarantees. In this work, we bound the KL divergence between the samples generated by our proximal diffusion algorithm and the true data distribution. Combined with existing theoretical results for proximal matching [23], our `ProxDM` provides provable guarantees for its generated samples.

### A.3 Bottleneck of Convergence at Large Step Numbers

Our results showed that the FID score of `ProxDM` deteriorates at large step numbers (see Fig. 3). We conjecture that this behavior arises because `ProxDM`, unlike score-based methods, relies on solving an implicit equation that in turn defines a proximal map. In practice, we use proximal matching to train networks to approximate the corresponding proximal operators, rather than implementing exact proximals at each step. As a result, the computed updates may violate the optimality conditions that define such updates, introducing inaccuracies that accumulate. Empirically, we observe that these inaccuracies appear more problematic than the approximation errors of scores in score-based samplers. Better network architectures and training strategies will provide more accurate and stable approximation of the proximal operators, leading to even higher empirical accuracy at high step counts.

## B  Notation

In this section, we introduce the notation we use and some necessary technical background.

**Glossary.**   Let us summarize the symbols used throughout the appendix:

- $h$: Discretization step size.
- $T$: Time horizon of forward process.
- $N$: Number of discretization steps $N = T/h$.
- $p_t$: Distribution of the forward process at time $t$.
- $\overleftarrow{p}_t$: Distribution of the reverse-time process at time $t$, i.e., $\overleftarrow{p}_t = p_{T-t}$.
- $q_t$: Distribution of samples from the algorithm at time $t$. The density of the samples after $k$ steps is $q_{kh}$. The density of the final samples is $q_T$. The density of the initial samples $q_0$ is a standard Gaussian.
- $\overleftarrow{p}_t^{(k)}$: Distribution of the reverse-time process within the $k$-th step of the algorithm, where $k \in \{0, 1, \dots, N-1\}$ and $t \in [0, h]$. That is $\overleftarrow{p}_t^{(k)} = \overleftarrow{p}_{kh+t} = p_{T-kh-t}$.
- $q_t^{(k)}$: Distribution of the interpolating process of the sampling algorithm within the $k$-th step (see Lemmas 1 and 2 and surrounding discussion). Note that $q_t^{(k)} = q_{kh+t}$.

**Remarks on terminology.**   We abuse notation by using the same symbols to denote distributions and their probability density function. We call the distribution we wish to sample from the target or data distribution. We use the word "process" to refer to a stochastic process.

**Linear algebra.**   The symbols $\mathbb{R}^d$, $\mathbb{R}^{d \times d}$, $\mathcal{S}^d$, and $\mathbb{R}^{d \times d \times d}$ denote the set of real $d$-dimensional vectors, $d \times d$ matrices, and $d \times d$ symmetric matrices, and $d \times d \times d$ tensors. We endow $\mathbb{R}^d$ with the standard inner product $\langle x, y \rangle = x^\top y$, which induces the $\ell_2$ norm via $\|x\| = \sqrt{\langle x, x \rangle}$. For a matrix $A \in \mathbb{R}^{d \times d}$ we let $\|A\|_{\mathrm{op}}$ denote the operator norm induced by the $\ell_2$ norm and $\|A\|_{\mathrm{F}}$ the Frobenius norm. In particular, if $A$ is symmetric with eigenvalues $\lambda(A) \in \mathbb{R}^d$, then $\|A\|_{\mathrm{op}} = \max_i |\lambda_i|$ and $\|A\|_{\mathrm{F}} = \|\lambda(A)\|$. Thus, $\|A\|_{\mathrm{F}} \leq \sqrt{d} \|A\|_{\mathrm{op}}$. For a symmetric matrix $Q \in \mathcal{S}^d$ with positive eigenvalues (i.e., a positive definite matrix), we write $\|x\|_Q = \sqrt{\langle x, Qx \rangle}$ for the associated norm.

**Calculus.**   Given a differentiable map $\phi \colon \mathbb{R}^d \to \mathbb{R}^k$, we use $\nabla \varphi(x)$ to denote its gradient or Jacobian at $x$ if $k = 1$ or $k > 1$, respectively. Similarly, given a $C^3$ function $\varphi \colon \mathbb{R}^d \to \mathbb{R}$, we use the symbols $\nabla^2 \varphi(x) \in \mathbb{R}^{d \times d}$ and $\nabla^3 \varphi(x)$ to denote its Hessian and tensor of third-order derivatives at $x$ whose components are given by

$$\nabla^2 \varphi(x)_{ij} = \frac{\partial^2 \varphi(x)}{\partial x_i \partial x_j} \quad \text{and} \quad \nabla^3 \varphi(x)_{ijk} = \frac{\partial^3 \varphi(x)}{\partial x_i \partial x_j \partial x_k},$$

respectively. We denote the Laplacian of $\varphi$ via

$$\Delta \varphi(x) = \mathrm{Tr}(\nabla^2 \varphi(x)) = \sum_{i=1}^d \frac{\partial^2 \varphi(x)}{\partial x_i^2} \in \mathbb{R}.$$

The symbol $\nabla\cdot$ denotes the divergence operator, which acts on a vector field $v\colon \mathbb{R}^d \to \mathbb{R}^d$ via

$$\nabla \cdot v(x) = \sum_{i=1}^{d} \frac{\partial v_i(x)}{\partial x_i} \in \mathbb{R}.$$

Note that with this notation in place we have the identity $\nabla \cdot (\nabla\varphi) = \Delta\varphi$. For a matrix-valued function $G\colon \mathbb{R}^d \to \mathbb{R}^{d\times d}$ and a differentiable function $\phi\colon \mathbb{R}^d \to \mathbb{R}$, let $\mathrm{Tr}(\nabla^3\varphi(x)G(x)) \in \mathbb{R}^d$ denote the vector whose $i$-th component is $\mathrm{Tr}(\nabla_i\nabla^2\varphi(x)G(x))$ where

$$\nabla_i\nabla^2\varphi(x) = \frac{\partial}{\partial x_i}\nabla^2\varphi(x) = \left(\frac{\partial^3\varphi(x)}{\partial x_i\partial x_j\partial x_k}\right)_{jk} \in \mathbb{R}^{d\times d}.$$

We use the symbol $\nabla \cdot G$ to denote the divergence operator broadcast to the rows of $G$, i.e.,

$$\nabla \cdot G(x) := [\nabla \cdot G_1(x), \ldots, \nabla \cdot G_d(x)]^\top \in \mathbb{R}^d$$

where $G_i(x) \in \mathbb{R}^d$ is the $i$-th row of $G(x)$. For brevity, we write $G$ in place of $G(x)$ when the argument is clear from context.

Finally, given a function $\varphi\colon \mathcal{X} \to \mathcal{Y}$ between finite dimensional vector spaces and a norm-one vector $v$ we define the directional derivative of $\varphi$ with respect to $v$ at the point $x$ as

$$\nabla_v\varphi(x) = \lim_{s\downarrow 0} \frac{\varphi(x + sv) - \varphi(x)}{u}.$$

Further, given a canonical basis $\{e_1, \ldots, e_{\dim\mathcal{X}}\}$ of $\mathcal{X}$ we let $\nabla_k\varphi$ be a shorthand for $\nabla_{e_k}\varphi$ for $k \in \{1, 2, ..., \dim\mathcal{X}\}$.

**Lipschitzness of the Hessian.**   For a function $f\colon \mathbb{R}^d \to \mathbb{R}$, we define

$$H_f(x) := \max\left\{\left\|\nabla_i\nabla^2 f(x)\right\|_{\mathrm{op}} \mid i = 1, 2, \ldots, d\right\} \in \mathbb{R}. \tag{10}$$

The condition $H_f(x) < \infty$ is weaker than a global Lipschitzness for the Hessian.[6] Indeed, if

$$\left\|\nabla^2 f(x) - \nabla^2 f(y)\right\|_{\mathrm{op}} \leq H\|x - y\| \qquad \text{for all } x, y \in \mathbb{R}^d,$$

then for all $x \in \mathbb{R}^d$,

$$\left\|\nabla_v\nabla^2 f(x)\right\|_{\mathrm{op}} \leq H \qquad \text{for all } v \in \mathbb{R}^d, \|v\| = 1,$$

which implies

$$\left\|\nabla_i\nabla^2 f(x)\right\|_{\mathrm{op}} \leq H \qquad \text{for } i = 1, 2, \ldots, d.$$

Comparing the above with (10), we can see that $H_f(x)$ is weaker than $H$ in two aspects: $(i)$ the quantity $H_f(x)$ is local, and $(ii)$ it only concerns the directional derivative with respect to canonical basis vectors $e_i$, not all directions. We use this definition because it suffices for our convergence analysis. We define the shorthand

$$H_t(x) := H_{\ln p_t}(x) \qquad \text{for } t \in \mathbb{R}. \tag{11}$$

where $p_t$ is the density along the forward process at time $t$.

## C   Formal statement of the convergence guarantee

Before introducing the formal statement, we sketch the proof strategy, which motivates some of the technical assumptions. We present the sketch here regarding the fully backward discretization, but the strategy for the hybrid method is analogous. This proof strategy is based on the analysis of Proximal Langevin Algorithm [79].

---

[6][79] assumes that the Hessian of the target potential has a global Lipschitz constant.

We consider the Ornstein–Uhlenbeck (OU) process as the forward process:

$$dX_t = -X_t dt + \sqrt{2}dW_t, \quad X_0 \sim p_0, \quad t \in [0, T], \tag{12}$$

and use $p_t$ to denote the distribution of $X_t$. For simplicity, we will consider reverse-time processes with time flowing forward. That is, although the original reverse time process evolves via

$$dX_t = [-X_t - 2\nabla \ln p_t(X_t)] \, dt + \sqrt{2}d\overline{W}_t \quad \text{where } t \text{ flows from } T \text{ to } 0 \tag{13}$$

and $\overline{W}_t$ is the backward Brownian motion, we consider the more intuitive

$$dX_t = [X_t + 2\nabla \ln p_{T-t}(X_t)] \, dt + \sqrt{2}dW_t \tag{14}$$

where $t$ flows from $0$ to $T$. We use $\overleftarrow{p}_t = p_{T-t}$ for the distribution of the process in (14) to distinguish it from that of the process in (13). We consider the backward discretization $\widehat{X}_{kh}$ evolving via the recursion

$$\widehat{X}_{(k+1)h} - \widehat{X}_{kh} = h\widehat{X}_{(k+1)h} + 2h\nabla \ln p_{T-(k+1)h}\left(\widehat{X}_{(k+1)h}\right) + \sqrt{2h}z_k, \tag{15}$$

here $k$ goes from $0$ to $N-1$, $z_k \sim \mathcal{N}(0, I_d)$ and $\widehat{X}_0 \sim \mathcal{N}(0, I_d)$. This is equivalent to `ProxDM` with the `hybrid` flag set to false (with the only difference that $k$ traverses from $0$ to $N-1$ as opposed to the reverse). When the `hybrid` flag is set to true, the discretization reads

$$\widehat{X}_{(k+1)h} - \widehat{X}_{kh} = h\widehat{X}_{kh} + 2h\nabla \ln p_{T-(k+1)h}\left(\widehat{X}_{(k+1)h}\right) + \sqrt{2h}z_k. \tag{16}$$

Our goal is to bound the KL divergence $\text{KL}(\overleftarrow{p}_T \, \| q_T)$ between $\overleftarrow{p}_T$, the target distribution (note that $\overleftarrow{p}_T = p_0$), and $q_T$, the distribution of $\widehat{X}_T$. If $\widehat{X}_T$ was defined by a continuous time process with sufficient regularity, then one could apply the fundamental theorem of calculus to get

$$\text{KL}(\overleftarrow{p}_T \, \| q_T) = \text{KL}(\overleftarrow{p}_0 \, \| q_0) + \int_0^T \frac{\partial \text{KL}(\overleftarrow{p}_t \, \| q_t)}{\partial t} dt. \tag{17}$$

In turn, bounding the derivative of the KL divergence as time evolves would yield a bound on the quantity of interest. To execute this plan in our discretized setting, we construct interpolating stochastic processes flowing from $\widehat{X}_{kh}$ to $\widehat{X}_{(k+1)h}$. Define the stochastic processes $\left(\widehat{X}_t^{(k)}\right)_{t \in [0,h]}$ given by

$$\widehat{X}_t^{(k)} = \widehat{X}_0^{(k)} + t\widehat{X}_t^{(k)} + 2t\nabla \ln p_{T-kh-t}\left(\widehat{X}_t^{(k)}\right) + \sqrt{2}W_t \tag{18}$$

starting at $\widehat{X}_0^{(k)} = \widehat{X}_{kh}$ for all $k$. The next lemma gives an explicit formula for the SDE governing the dynamics of this stochastic process; we defer its proof to Appendix D.1.

**Lemma 1** (Interpolating Process for Backward Algorithm). *Suppose that $h \leq \frac{1}{2L+1}$. Then, the discrete algorithm in (15) has a continuous interpolating process evolving according to the SDE*

$$d\widehat{X}_t^{(k)} = \widehat{\mu}_k(\widehat{X}_t^{(k)}, t) \, dt + \sqrt{2\widehat{G}_k\left(\widehat{X}_t^{(k)}, t\right)} \, dW_t \qquad t \in [0, h] \tag{19}$$

*where*

$$\widehat{G}_k(x, t) = \left[(1-t)I_d - 2t\nabla^2 \ln p_{T-kh-t}(x)\right]^{-2}, \quad \text{and}$$

$$\widehat{\mu}_k(x, t) = \sqrt{\widehat{G}_k(x, t)} \left(x + 2\nabla \ln p_{T-kh-t}(x) + 2t\frac{\partial}{\partial t}\nabla \ln p_{T-kh-t}(x)\right.$$

$$\left. + 2t \, \text{Tr}\left[\nabla^3 \ln p_{T-kh-t}(x)\widehat{G}_k(x, t)\right]\right).$$

We shall compare the distributions of these processes with the equivalent pieces of (14), i.e.,

$$dX_t^{(k)} = \underbrace{\left[X_t^{(k)} + 2\nabla \ln p_{T-kh-t}\left(X_t^{(k)}\right)\right]}_{:=\mu_k\left(X_t^{(k)}, t\right)} dt + \sqrt{2} \underbrace{I}_{:=G_k\left(X_t^{(k)}, t\right)} dW_t \qquad t \in [0, h] \tag{20}$$

To derive a bound on the time derivative of KL divergence in (17), we leverage the Fokker-Planck equation in tandem with a boundary condition that allows us to apply an integral by parts. This motivates the assumptions we impose next.

**Assumption 1** (**Regularity of the potential**). *The following hold.*

*(**Smoothness**) There exists $L > 1$, such that for all $t \geq 0$, $\ln p_t$ is three-times differentiable and $L$-smooth, i.e. the gradient $\nabla \ln p_t$ is $L$-Lipschitz.*

*(**Data second moment**) The expected energy $M_2 := \mathbb{E}_{p_0}[\|x\|^2] < \infty$ is finite.*

*(**Score second moment**) The running average $C_p := \frac{1}{T} \int_0^T \mathbb{E}_{p_t}[\|\nabla \ln p_t(x)\|^2] dt < \infty$ is finite.*

*(**Lipschitz Hessian**) For any $x$ and $t$, let $H_t(x) = \max\left\{ \left\|\nabla_i \nabla^2 \ln p_t(x)\right\|_{\mathrm{op}} \mid i = 1, 2, \ldots, d\right\}$. The running average $H^2 := \frac{1}{T} \int_0^T \mathbb{E}_{p_t}[H_t(x)^2] dt < \infty$ is finite.*

We highlight that we do not require the distribution to be log-concave, nor do we impose that it satisfies the log-Sobolev inequality. Smoothness of the potential and bounded second moments are standard assumptions in the literature [6, 81]. Lipschitz continuity of the Hessian is not as common for diffusion models, but has been used to study Langevin-type methods [54, 79].

Additionally, we assume regularity conditions for the analysis of time derivative of KL. We first state it in its general form.

**Assumption 2** (**Regularity of KL time derivative**). *Consider the SDEs*

$$dY_t = f(Y_t, t)dt + \sqrt{2J(Y_t, t)}dW_t$$

*and*

$$d\widehat{Y}_t = \widehat{f}\left(\widehat{Y}_t, t\right)dt + \sqrt{2\widehat{J}\left(\widehat{Y}_t, t\right)}dW_t$$

*where $Y_t, \widehat{Y}_t \in \mathbb{R}^d$ and $f, \widehat{f}\colon \mathbb{R}^d \times \mathbb{R} \to \mathbb{R}^d$, and $J, \widehat{J}\colon \mathbb{R}^d \times \mathbb{R} \to \mathcal{S}_+^d$.[7] Let $\rho_t$ and $\nu_t$ be the distributions of $Y_t$ and $\widehat{Y}_t$ respectively. The following two conditions hold.*

*(**Boundary condition**) For all $t$,*

$$\int_{\mathbb{R}^d} \nabla \cdot \left\{ \ln \frac{\rho_t}{\nu_t} \left[\nabla \cdot (\rho_t J) - \rho_t f\right] \right\} dx = 0$$

$$\text{and} \quad \int_{\mathbb{R}^d} \nabla \cdot \left\{ \frac{\rho_t}{\nu_t} \left[\nabla \cdot \left(\nu_t \widehat{J}\right) - \nu_t \widehat{f}\right] \right\} dx = 0.$$

*Here, we write $\rho_t$, $\nu_t$, $f$, $\widehat{f}$, $J$ and $\widehat{J}$ in place of $\rho_t(x)$, $\nu_t(x)$, $f(x, t)$, $\widehat{f}(x, t)$, $J(x, t)$ and $\widehat{J}(x, t)$, respectively.*

*(**Dominated convergence**) There exist two integrable functions $\theta, \kappa\colon \mathbb{R}^d \to \mathbb{R}$ such that*

$$\left| \frac{\partial}{\partial t} \left( \rho_t(x) \ln \frac{\rho_t(x)}{\nu_t(x)} \right) \right| \leq \theta(x) \quad \text{and} \quad \left| \frac{\partial}{\partial t} \rho_t(x) \right| \leq \kappa(x)$$

*for all $t$ and almost every $x$.[8]*

To analyze the backward algorithm (`ProxDM` with `hybrid` flag set to false), we instantiate Assumption 2 for the reverse-time and interpolating processes.

**Assumption 3** (**Regularity Condition for Interpolating Process of Backward Algorithm**). *For each $k \in \{0, \ldots, N-1\}$, let $\mu_k$, $G_k$, and $X_t^{(k)}$ be defined as in (20) and let $\overleftarrow{p}_t^{(k)}$ be the density of $X_t^{(k)}$. Similarly, let $\widehat{\mu}_k$, $\widehat{G}_k$, and $\widehat{X}_t^{(k)}$ be defined as in (19) and let $q_t^{(k)}$ be the density of $\widehat{X}_t^{(k)}$. Assumption 2 holds for $f = \mu_k, \widehat{f} = \widehat{\mu}_k, J = G_k, \widehat{J} = \widehat{G}_k, \rho_t = \overleftarrow{p}_t^{(k)}$, and $\nu_t = q_t^{(k)}$.*

The boundary condition is required for applying an integral by parts in the analysis of time derivative of KL (see Appendix C.2). Similar conditions are implicitly assumed in proofs in prior works [6, 79, 41, 78]. These conditions are met when the functions inside the divergence operator decay

---

[7]Recall that $\mathcal{S}_+^d$ denotes the set of $d \times d$ symmetric positive semidefinite matrices.
[8]By integrable we mean $\int \theta(x) dx < \infty$ and $\int \kappa(x) dx < \infty$.

sufficiently fast at infinity, which is satisfied by distributions with regular tail behaviors. The domination condition is required to apply the Leibniz integral rule in the analysis of time derivative of KL (see Lemma 20).

To state an analogous assumption for the hybrid algorithm we require a slightly different interpolating SDE, which we introduce in the next result. The proof of this lemma is deferred to Appendix D.2.

**Lemma 2** (Interpolating Process for Hybrid Algorithm). *Suppose that* $h < \frac{1}{2L}$. *Then, the discrete algorithm in (16) has a continuous interpolating process evolving according to the SDE*

$$d\widehat{X}_t^{(k)} = \widehat{\mu}_k(\widehat{X}_t^{(k)}, t; \widehat{X}_0^{(k)}) \, dt + \sqrt{2\widehat{G}_k\left(\widehat{X}_t^{(k)}, t\right)} \, dW_t \qquad t \in [0, h] \tag{21}$$

*where*

$$\widehat{G}_k(x, t) = \left[I_d - 2t\nabla^2 \ln p_{T-kh-t}(x)\right]^{-2}, \quad and$$

$$\widehat{\mu}_k(x, t; a) = \sqrt{\widehat{G}_k(x, t)} \left(a + 2\nabla \ln p_{T-kh-t}(x) + 2t\frac{\partial}{\partial t}\nabla \ln p_{T-kh-t}(x)\right.$$

$$\left. + 2t \operatorname{Tr}\left[\nabla^3 \ln p_{T-kh-t}(x)\widehat{G}_k(x, t)\right]\right).$$

With this hybrid interpolating process in place, we introduce analogous regularity conditions.

**Assumption 4** (**Regularity Condition for Interpolating Process of Hybrid Algorithm**). *For each* $k \in \{0, \dots, N-1\}$, *let* $\mu_k$, $G_k$, *and* $X_t^{(k)}$ *be defined as in (20) and let* $\overleftarrow{p}_{t|0}^{(k)}(\cdot \mid a)$ *denote the conditional density of* $X_t^{(k)}$ *given* $X_0^{(k)} = a$. *Similarly, let* $\widehat{\mu}_k$, $\widehat{G}_k$, *and* $\widehat{X}_t^{(k)}$ *be defined as in (21) and let* $q_{t|0}^{(k)}(\cdot \mid a)$ *be the conditional density of* $\widehat{X}_t^{(k)}$ *given* $\widehat{X}_0^{(k)} = a$. *Assumption 2 holds for* $f = \mu_k$, $\widehat{f}(x, t) = \widehat{\mu}_k(x, t; a)$, $J = G_k$, $\widehat{J} = \widehat{G}_k$, $\rho_t = \overleftarrow{p}_{t|0}^{(k)}(\cdot \mid a)$ *and* $\nu_t = q_{t|0}^{(k)}(\cdot \mid a)$.

Equipped with these assumptions, we are ready to state our main result.

**Theorem 2** (**Convergence guarantee**). *Suppose that Assumption 1 holds. Set the step size and time horizon to satisfy* $h \leq \frac{1}{8L+4}$ *and* $T \geq 0.25$. *The following two hold true.*

*(**Hybrid**) If the* `hybrid` *flag is set to true and Assumption 4 holds, then*

$$\operatorname{KL}\left(\overleftarrow{p}_T \| q_T\right) \lesssim (d + M_2)e^{-T} + hTdL^2 + h^2T\left[(d + M_2 + C_p)L^2 + d^2H^2\right] + h^4Td^3L^2H^2. \tag{22}$$

*(**Backward**) If the* `hybrid` *flag is set to false and Assumption 3 holds, then*

$$\operatorname{KL}\left(\overleftarrow{p}_T \| q_T\right) \lesssim (d + M_2)e^{-T} + h^2T\left[(d + M_2 + C_p)L^2 + d^2H^2\right] + h^4Td^3L^2H^2. \tag{23}$$

*Proof.* The proof strategies for both hybrid and backward are completely analogous and follow from the identity (17). The two proofs diverge slightly in how we upper bound the derivative of the KL divergence. The core of the argument appears in the next proposition, whose proof is deferred to Appendix C.1.

**Proposition 1.** *The following two hold for any* $k \in \{0, \dots, N-1\}$.

*(**Hybrid**) If the* `hybrid` *flag is set to true and Assumption 4 holds, then*

$$\operatorname{KL}\left(\overleftarrow{p}_h^{(k)} \| q_h^{(k)}\right) - \operatorname{KL}\left(\overleftarrow{p}_0^{(k)} \| q_0^{(k)}\right) \lesssim h^2dL^2 + h^2L^2 \int_0^h \mathbb{E}_{\overleftarrow{p}_t^{(k)}}\left[\|x\|^2 + \|\nabla \ln \overleftarrow{p}_t^{(k)}\|^2\right] dt$$

$$+ \left(h^2d^2 + h^4d^3L^2\right) \int_0^h \mathbb{E}_{\overleftarrow{p}_t^{(k)}}\left[H_{T-kh-t}(x)^2\right] dt. \tag{24}$$

*(**Backward**) If the `hybrid` flag is set to false and Assumption 3 holds, then*

$$\text{KL}\left(\overleftarrow{p}_h^{(k)} \,\|\, q_h^{(k)}\right) - \text{KL}\left(\overleftarrow{p}_0^{(k)} \,\|\, q_0^{(k)}\right) \lesssim h^2 L^2 \int_0^h \mathbb{E}_{\overleftarrow{p}_t^{(k)}}\left[\|x\|^2 + \|\nabla \ln \overleftarrow{p}_t^{(k)}\|^2\right] dt$$

$$+ \left(h^2 d^2 + h^4 d^3 L^2\right) \int_0^h \mathbb{E}_{\overleftarrow{p}_t^{(k)}}\left[H_{T-kh-t}(x)^2\right] dt. \tag{25}$$

The main difference arises from the first term in the upper bound of the hybrid case. From now on, we will continue only with the hybrid scenario, as the rest of the argument is analogous to purely backward discretization. Taking the sum of the inequalities in Proposition 1 from 0 to $N-1$ yields

$$\text{KL}\left(\overleftarrow{p}_T \,\|\, q_T\right) - \text{KL}\left(\overleftarrow{p}_0 \,\|\, q_0\right) = \sum_{k=0}^{N-1}\left(\text{KL}\left(\overleftarrow{p}_h^{(k)} \,\|\, q_h^{(k)}\right) - \text{KL}\left(\overleftarrow{p}_0^{(k)} \,\|\, q_0^{(k)}\right)\right)$$

$$\lesssim N h^2 d L^2 + h^2 L^2 \int_0^T \mathbb{E}_{\overleftarrow{p}_t}\left[\|x\|^2 + \|\nabla \ln \overleftarrow{p}_t\|^2\right] dt \tag{26}$$

$$+ \left(h^2 d^2 + h^4 d^3 L^2\right) \int_0^T \mathbb{E}_{\overleftarrow{p}_t}\left[H_t(x)^2\right] dt.$$

To finish the argument, we invoke the following two results, whose proofs are standard, but we include them for completeness in Appendix D.3 and Appendix D.4, respectively.

**Lemma 3.** *The second moment of $\overleftarrow{p}_t$ is bounded*

$$\mathbb{E}_{\overleftarrow{p}_t}[\|x\|^2] \le 2(d + M_2) \quad \text{for all } t \in [0, T].$$

**Lemma 4** (Lemma 9 in [6])**.** *Let $q_0$ be the standard normal distribution in $d$ dimensions. For $T \ge 0.25$, we have*

$$\text{KL}(\overleftarrow{p}_0 \,\|\, q_0) \le 2(d + M_2)e^{-2T}.$$

Substituting these two into (26), using $N = T/h$ and Assumption 1 and reordering yields

$$\text{KL}\left(\overleftarrow{p}_T \,\|\, q_T\right) \lesssim (d + M_2)e^{-2T} + h T d L^2 + h^2 T L^2 \left(d + M_2 + C_p\right) + \left(h^2 d^2 + h^4 d^3 L^2\right) T H^2.$$

Thus, (22) follows. The argument for (23) is analogous, which completes the proof. $\qquad\square$

The rest of this section is devoted to proving Proposition 1.

## C.1 Proof of Proposition 1

Once more, the goal is to use the identity (17) by bounding the time derivative of the KL discrepancy. The crux of this proof lies in the following lemma, which we will instantiate twice, once for the hybrid discretization and once for the backward discretization. We defer its proof to the end of this section (Appendix C.2).

**Lemma 5** (Time derivative of KL between SDEs)**.** *Suppose Assumption 2 holds.*

*Then, for all $t > 0$ we have*

$$\frac{d}{dt}\text{KL}(\rho_t \| \nu_t) = -\mathbb{E}_{\rho_t}\left[\left\|\nabla \ln \frac{\rho_t}{\nu_t}\right\|_{\widehat{J}}^2\right] + \mathbb{E}_{\rho_t}\left[\left\langle f - \widehat{f}, \nabla \ln \frac{\rho_t}{\nu_t}\right\rangle\right]$$

$$+ \mathbb{E}_{\rho_t}\left[\left\langle (\widehat{J} - J)\nabla \ln \rho_t + \nabla \cdot (\widehat{J} - J), \nabla \ln \frac{\rho_t}{\nu_t}\right\rangle\right] \tag{27}$$

*where the arguments for $f$, $\widehat{f}$, $J$ and $\widehat{J}$ are omitted.*

We consider two cases. We start with the fully backward discretization since the argument is slightly more straightforward.

**Case 1:** Suppose that `hybrid` flag is set to false. Fix $k \in \{0, \dots, N-1\}$. Let $\widehat{X}_t^{(k)}$, $\widehat{\mu}_k$ and $\widehat{G}_k$ defined as in Lemma 1. First, we state some properties of $\widehat{G}_k$, with their proof deferred to Appendix D.5.

**Lemma 6.** *Let $\widehat{G}_k$ defined as in Lemma 1. Then, we have that*

$$(1 - 4tL)I \preceq \widehat{G}_k \preceq (1 + 18tL)I \quad and \quad (1 - 2tL)I \preceq \sqrt{\widehat{G}_k} \preceq (1 + 6tL)I.$$

We instantiate Lemma 5 with $f = \mu_k$, $\widehat{f} = \widehat{\mu}_k$, $J = G_k$, and $\widehat{J} = \widehat{G}_k$. In which case, $\rho_t = \overleftarrow{p}_t^{(k)}$ and $\nu_t = q_t^{(k)}$ and the lemma yields

$$
\frac{d}{dt} \mathrm{KL}(\overleftarrow{p}_t^{(k)} \| q_t^{(k)})
$$

$$
= -\mathbb{E}_{\overleftarrow{p}_t^{(k)}} \left[ \left\| \nabla \ln \frac{\overleftarrow{p}_t^{(k)}}{q_t^{(k)}} \right\|_{\widehat{G}_k}^2 \right] + \mathbb{E}_{\overleftarrow{p}_t^{(k)}} \left[ \left\langle \mu_k - \widehat{\mu}_k, \nabla \ln \frac{\overleftarrow{p}_t^{(k)}}{q_t^{(k)}} \right\rangle \right]
$$

$$
+ \mathbb{E}_{\overleftarrow{p}_t^{(k)}} \left[ \left\langle (\widehat{G}_k - I) \nabla \ln \overleftarrow{p}_t^{(k)} + \nabla \cdot (\widehat{G}_k - I), \nabla \ln \frac{\overleftarrow{p}_t^{(k)}}{q_t^{(k)}} \right\rangle \right]
$$

$$
\leq -\frac{1}{2} \mathbb{E}_{\overleftarrow{p}_t^{(k)}} \left[ \left\| \nabla \ln \frac{\overleftarrow{p}_t^{(k)}}{q_t^{(k)}} \right\|^2 \right] + 2\mathbb{E}_{\overleftarrow{p}_t^{(k)}} \left[ \|\mu - \widehat{\mu}\|^2 \right] + 2\mathbb{E}_{\overleftarrow{p}_t^{(k)}} \left[ \left\| (\widehat{G}_k - I) \nabla \ln \overleftarrow{p}_t^{(k)} \right\|^2 \right]
$$

$$
+ 2\mathbb{E}_{\overleftarrow{p}_t^{(k)}} \left[ \left\| \nabla \cdot (\widehat{G}_k - I) \right\|^2 \right] + \frac{3}{8} \mathbb{E}_{\overleftarrow{p}_t^{(k)}} \left[ \left\| \nabla \ln \frac{\overleftarrow{p}_t^{(k)}}{q_t^{(k)}} \right\|^2 \right]
$$

$$
\leq 2\mathbb{E}_{\overleftarrow{p}_t^{(k)}} \left[ \|\mu - \widehat{\mu}\|^2 + \left\| (\widehat{G}_k - I) \nabla \ln \overleftarrow{p}_t^{(k)} \right\|^2 + \left\| \nabla \cdot \widehat{G}_k \right\|^2 \right] \tag{28}
$$

where the second relation uses Cauchy-Schwarz and Young's inequalities to bound $\langle a, b \rangle \leq 2\|a\|^2 + \frac{1}{8}\|b\|^2$ and the fact that $\frac{1}{2}\| \cdot \|^2 \leq \| \cdot \|_{\widehat{G}_k}^2$ since $\frac{1}{2}I \preceq \widehat{G}_k$ thanks to Lemma 6 and the constraint on $t \leq h \leq \frac{1}{8L}$. The following two lemmas will help us bound the first and last terms in the expected value in (28). Their proofs are deferred to Appendices D.7 and D.9, respectively.

**Lemma 7.** *For all $k \in \{0, \dots, N-1\}$, the difference between $\mu_k$ and $\widehat{\mu}_k$ is bounded by*

$$
\|\mu_k - \widehat{\mu}_k\|^2 \leq t^2 L^2 \left( 363\, \|x\|^2 + 2187\, \|\nabla \ln \overleftarrow{p}_t^{(k)}(x)\|^2 \right) + 24300\, t^4 d^3 L^2 H_{T-kh-t}(x)^2.
$$

**Lemma 8.** *For all $k \in \{0, \dots, N-1\}$, the norm of $\nabla \cdot \widehat{G}_k$ is bounded by*

$$
\|\nabla \cdot \widehat{G}_k(x)\| \leq 364\, tdH_{T-kh-t}(x).
$$

Furthermore, Lemma 6 gives $\|\widehat{G}_k - I\|_{\mathrm{op}}^2 \leq (18tL)^2 = 324t^2L^2$. Substituting all of these bounds into (28) gives

$$
\frac{d}{dt} \mathrm{KL}(\overleftarrow{p}_t^{(k)} \| q_t^{(k)}) \lesssim \mathbb{E}_{\overleftarrow{p}_t^{(k)}} \left[ t^2 L^2 (\|x\|^2 + \|\nabla \ln \overleftarrow{p}_t^{(k)}(x)\|^2) + \left( t^2 d^2 + t^4 d^3 L^2 H_{T-kh-t}(x)^2 \right) \right].
$$

Upper bounding $t \leq h$ and integrating from $t = 0$ to $t = h$ yields the desired inequality (25); finishing the proof for the fully backward discretization.

**Case 2:** Suppose that `hybrid` flag is set to true. Let $\widehat{X}_t^{(k)}$, $\widehat{\mu}_k$ and $\widehat{G}_k$ defined as in Lemma 2. The proof of this case is slightly different: we use the following chain rule [33, 6], before applying the fundamental theorem of calculus

$$
\mathrm{KL}\left(\overleftarrow{p}_h^{(k)} \| q_h^{(k)}\right) - \mathrm{KL}\left(\overleftarrow{p}_0^{(k)} \| q_0^{(k)}\right) \leq \mathbb{E}_{a \sim \overleftarrow{p}_0^{(k)}} \left[ \mathrm{KL}\left(\overleftarrow{p}_{h|0}^{(k)}(\cdot \mid a) \| q_{h|0}^{(k)}(\cdot \mid a)\right) \right] \tag{29}
$$

where $q_{t|0}^{(k)}(\cdot \mid a)$ denote the conditional density of $\widehat{X}_t^{(k)}$ given $\widehat{X}_0^{(k)} = a$, and $\overleftarrow{p}_{t|0}^{(k)}(\cdot \mid a)$ denote the conditional density of $X_t^{(k)}$ in (20) given $X_0^{(k)} = a$. Notice that we can express the KL divergence

on the right-hand side by applying the fundamental theorem of calculus in a similar fashion as in (17). However, in this case the value at $t = 0$ satisfies $\mathrm{KL}\left(\overleftarrow{p}_{0|0}^{(k)}\left(\cdot \mid a\right) \| q_{0|0}^{(k)}(\cdot \mid a)\right) = \mathrm{KL}\left(\delta_a \| \delta_a\right) = 0$. Thus, the right-hand side is bounded by $\mathbb{E}_{a \sim \overleftarrow{p}_0^{(k)}}\left[\int_0^h \frac{d}{dt}\mathrm{KL}\left(\overleftarrow{p}_{t|0}^{(k)}\left(\cdot \mid a\right) \| q_{t|0}^{(k)}(\cdot \mid a)\right) dt\right]$. Next, we bound the time derivative. Paralleling the proof in **Case 1**, we introduce relevant properties of $\widehat{G}_k$; their proof is deferred to Appendix D.6.

**Lemma 9.** *Let $\widehat{G}_k$ be defined as in Lemma 2. Then, we have that*

$$(1 - 4tL)I \preceq \widehat{G}_k \preceq (1 + 12tL)I \quad and \quad (1 - 2tL)I \preceq \sqrt{\widehat{G}_k} \preceq (1 + 4tL)I.$$

We invoke Lemma 5 with $f = \mu_k$, $\widehat{f}(x,t) = \widehat{\mu}_k(x,t;a)$, $J = G_k$, $\widehat{J} = \widehat{G}_k$, $\rho_t = \overleftarrow{p}_{t|0}^{(k)}\left(\cdot \mid a\right)$ and $\nu_t = q_{t|0}^{(k)}(\cdot \mid a)$. The lemma yields

$$\frac{d}{dt}\mathrm{KL}\left(\overleftarrow{p}_{t|0}^{(k)}\left(\cdot \mid a\right) \| q_{t|0}^{(k)}(\cdot \mid a)\right)$$

$$= -\mathbb{E}_{\overleftarrow{p}_{t|0}^{(k)}(\cdot|a)}\left[\left\|\nabla \ln \frac{\overleftarrow{p}_{t|0}^{(k)}\left(\cdot \mid a\right)}{q_{t|0}^{(k)}(\cdot \mid a)}\right\|_{\widehat{G}_k}^2\right]$$

$$+ \mathbb{E}_{\overleftarrow{p}_{t|0}^{(k)}(\cdot|a)}\left[\left\langle \mu_k - \widehat{\mu}_k, \nabla \ln \frac{\overleftarrow{p}_{t|0}^{(k)}\left(\cdot \mid a\right)}{q_{t|0}^{(k)}(\cdot \mid a)}\right\rangle\right]$$

$$+ \mathbb{E}_{\overleftarrow{p}_{t|0}^{(k)}(\cdot|a)}\left[\left\langle \left(\widehat{G}_k - G_k\right)\nabla \ln \overleftarrow{p}_{t|0}^{(k)}\left(\cdot \mid a\right) + \nabla \cdot \left(\widehat{G}_k - G_k\right), \nabla \ln \frac{\overleftarrow{p}_{t|0}^{(k)}\left(\cdot \mid a\right)}{q_{t|0}^{(k)}(\cdot \mid a)}\right\rangle\right].$$

Applying $\frac{1}{2}I \preceq \widehat{G}_k$—thanks to Lemma 9—in tandem with the analogous argument we used for the purely backward method (28) we derive

$$\frac{d}{dt}\mathrm{KL}\left(\overleftarrow{p}_{t|0}^{(k)}\left(\cdot \mid a\right) \| q_{t|0}^{(k)}(\cdot \mid a)\right)$$

$$\leq 2\mathbb{E}_{\overleftarrow{p}_{t|0}^{(k)}(\cdot|a)}\left[\|\mu_k - \widehat{\mu}_k\|^2 + \left\|\left(\widehat{G}_k - I\right)\nabla \ln \overleftarrow{p}_{t|0}^{(k)}\left(\cdot \mid a\right)\right\|^2 + \left\|\nabla \cdot \widehat{G}_k\right\|^2\right].$$

Integrating over $t \in [0, h]$ and taking expectation over $a \sim \overleftarrow{p}_0^{(k)}$, we have

$$\mathbb{E}_{a \sim \overleftarrow{p}_0^{(k)}}\left[\mathrm{KL}\left(\overleftarrow{p}_{h|0}^{(k)}\left(\cdot \mid a\right) \| q_{h|0}^{(k)}(\cdot \mid a)\right)\right]$$

$$\leq 2\int_0^h \mathbb{E}_{a \sim \overleftarrow{p}_0^{(k)}}\mathbb{E}_{x \sim \overleftarrow{p}_{t|0}^{(k)}(\cdot|a)}\left[\|\mu_k - \widehat{\mu}_k\|^2 + \left\|\left(\widehat{G}_k - I\right)\nabla \ln \overleftarrow{p}_{t|0}^{(k)}\left(x \mid a\right)\right\|^2 + \left\|\nabla \cdot \widehat{G}_k\right\|^2\right] dt.$$
$$\tag{30}$$

The following two lemmas will help us bound the first and last terms in the expected value above. Their proofs are deferred to Appendices D.8 and D.10, respectively.

**Lemma 10.** *For all $k \in \{0, \ldots, N-1\}$, the difference between $\mu_k$ and $\widehat{\mu}_k$ is bounded by*

$$\|\mu_k - \widehat{\mu}_k\| \leq tL\left(8\|x\| + 20\|\nabla \ln p_{T-kh-t}(x)\|\right) + 48\,t^2 d^{3/2}LH_{T-kh-t}(x) + 2\|a - x\|.$$

**Lemma 11.** *For all $k \in \{0, \ldots, N-1\}$, the norm of $\nabla \cdot \widehat{G}_k$ is bounded by*

$$\left\|\nabla \cdot \widehat{G}_k(x)\right\| \leq 192\,tdH_{T-kh-t}(x).$$

Furthermore, Lemma 9 gives that $\left\|\widehat{G}_k - I\right\|_{\mathrm{op}} \leq 12tL$. Substituting all of these bounds into (30) gives

$$\mathbb{E}_{a\sim\overleftarrow{p}_0^{(k)}}\left[\mathrm{KL}\left(\overleftarrow{p}_{h|0}^{(k)}\left(\cdot\mid a\right)\,\|\,q_{h|0}^{(k)}(\cdot\mid a)\right)\right]$$

$$\lesssim \int_0^h t^2 L^2 \mathbb{E}_{x\sim\overleftarrow{p}_t^{(k)}}\left[\|x\|^2 + \|\nabla\ln p_{T-kh-t}(x)\|^2\right]dt$$

$$+ \int_0^h (t^4 d^3 L^2 + t^2 d^2)\mathbb{E}_{x\sim\overleftarrow{p}_t^{(k)}}\left[H_{T-kh-t}(x)^2\right]dt$$

$$+ \int_0^h \mathbb{E}_{(a,x)\sim\left(\overleftarrow{p}_0^{(k)},\overleftarrow{p}_t^{(k)}\right)}\left[\|a-x\|^2\right]dt$$

$$+ \int_0^h t^2 L^2 \mathbb{E}_{(a,x)\sim\left(\overleftarrow{p}_0^{(k)},\overleftarrow{p}_t^{(k)}\right)}\left[\nabla\ln\overleftarrow{p}_{t|0}^{(k)}\left(x\mid a\right)\right]dt. \tag{31}$$

We now establish two lemmas that bound the last two terms on the right hand side above. Lemma 12 is adapted from [6, Lemma 10]. We defer the proof of Lemma 13 to Appendix D.11.

**Lemma 12.** *For all $k \in \{0, \ldots, N-1\}$ we have*

$$\mathbb{E}_{(a,x)\sim\left(\overleftarrow{p}_0^{(k)},\overleftarrow{p}_t^{(k)}\right)}\left[\|a-x\|^2\right] \leq 4td + 2t^2\mathbb{E}_{x\sim\overleftarrow{p}_t^{(k)}}\left[\|x\|^2\right].$$

*Proof of Lemma 12.* For ease of notation, we let $\nu_0 = \overleftarrow{p}_0^{(k)}$, $\nu_t = \overleftarrow{p}_t^{(k)}$, and $\nu_{0|t} = \overleftarrow{p}_{0|t}^{(k)}$ within this proof. Applying the triangle inequality and $(a+b)^2 \leq 2a^2 + 2b^2$ yields

$$\mathbb{E}_{(a,x)\sim(\nu_0,\nu_t)}\left[\|a-x\|^2\right] \leq 2\mathbb{E}_{x\sim\nu_t}\mathbb{E}_{a\sim\nu_{0|t}(\cdot|x)}\left[\|a-e^{-t}x\|^2\right] + 2\mathbb{E}_{x\sim\nu_t}\left[\|e^{-t}x-x\|^2\right].$$

Next we bound each term in this sum. Since $\nu_{0|t}(\cdot\mid x) = \mathcal{N}(e^{-t}x, (1-e^{-2t})I)$, we have

$$\mathbb{E}_{x\sim\nu_t}\mathbb{E}_{a\sim\nu_{0|t}(\cdot|x)}\left[\|a-e^{-t}x\|^2\right] = d(1-e^{-2t}) \leq 2td.$$

For the second term, factorizing and using the fact that $(1-e^{-t})^2 \leq t^2$ for all $t \geq 0$ gives

$$\mathbb{E}_{x\sim\nu_t}\left[\|e^{-t}x-x\|^2\right] = (1-e^{-t})^2\,\mathbb{E}_{x\sim\nu_t}\left[\|x\|^2\right] \leq t^2\,\mathbb{E}_{x\sim\nu_t}\left[\|x\|^2\right].$$

Combining the previous three bounds finishes the proof of Lemma 12. $\qquad\square$

**Lemma 13.** *For all $k \in \{0, \ldots, N-1\}$ we have*

$$t^2 L^2\,\mathbb{E}_{(a,x)\sim\left(\overleftarrow{p}_0^{(k)},\overleftarrow{p}_t^{(k)}\right)}\left[\|\nabla\ln\overleftarrow{p}_{t|0}^{(k)}\left(x\mid a\right)\|^2\right] \lesssim tdL^2 + t^2 L^2\mathbb{E}_{x\sim\overleftarrow{p}_t^{(k)}}\left[\|\nabla\ln p_{T-kh-t}(x)\|^2\right].$$

Applying Lemmas 12 and 13 into (31), using the chain rule (29), and recalling that $t \leq h$ and $L \geq 1$ yields inequality (24), finishing the proof of **Case 2**.

This concludes the proof of Proposition 1. $\qquad\square$

## C.2  Proof of Lemma 5

*Proof.* By the Fokker-Planck equation,

$$\frac{\partial\nu_t}{\partial t} = \nabla\cdot\left[\nabla\cdot\left(\nu_t\widehat{J}\right) - \nu_t\widehat{f}\right],$$

$$\frac{\partial\rho_t}{\partial t} = \nabla\cdot\left[\nabla\cdot\left(\rho_t J\right) - \rho_t f\right].$$

We now use a result on the time derivative of KL from prior work [78, Section 8.1.2]; see also [79, 6]. For completeness, we provide a full statement of the result in Lemma 20 and a proof in Appendix D.12, which relies the domination conditions. By Lemma 20, the time derivative of KL satisfies

$$\frac{d}{dt}\mathrm{KL}(\rho_t\|\nu_t) = \int\left(\frac{\partial\rho_t}{\partial t}\ln\frac{\rho_t}{\nu_t} - \frac{\partial\nu_t}{\partial t}\frac{\rho_t}{\nu_t}\right)dx.$$

Plugging in the Fokker-Planck equations,

$$\frac{d}{dt}\mathrm{KL}(\rho_t\|\nu_t) = \int \left\{\nabla\cdot[\nabla\cdot(\rho_t J) - \rho_t f]\ln\frac{\rho_t}{\nu_t} - \nabla\cdot\left[\nabla\cdot\left(\nu_t\widehat{J}\right) - \nu_t\widehat{f}\right]\frac{\rho_t}{\nu_t}\right\}dx$$

$$= \int\left\{-\left\langle\nabla\cdot(\rho_t J) - \rho_t f, \nabla\ln\frac{\rho_t}{\nu_t}\right\rangle + \left\langle\nabla\cdot(\nu_t\widehat{J}) - \nu_t\widehat{f}, \nabla\frac{\rho_t}{\nu_t}\right\rangle\right\}dx \quad (32)$$

where in the second equality we used integration by parts, which holds because of the boundary conditions. In the rest of the proof, we will transform the integrand in the equation above to show the desired result. For better readability, we omit the subscript $t$ in $\rho_t$ and $\nu_t$ from this point onward.

For the integrand, we have

$$-\left\langle\nabla\cdot(\rho J) - \rho f, \nabla\ln\frac{\rho}{\nu}\right\rangle + \left\langle\nabla\cdot(\nu\widehat{J}) - \nu\widehat{f}, \nabla\frac{\rho}{\nu}\right\rangle$$

$$= -\left\langle\nabla\cdot(\rho J) - \rho f, \nabla\ln\frac{\rho}{\nu}\right\rangle + \left\langle\nabla\cdot(\nu\widehat{J}) - \nu\widehat{f}, \frac{\rho}{\nu}\nabla\ln\frac{\rho}{\nu}\right\rangle$$

$$= -\left\langle\nabla\cdot(\rho J) - \rho f, \nabla\ln\frac{\rho}{\nu}\right\rangle + \left\langle\frac{\rho}{\nu}\nabla\cdot(\nu\widehat{J}) - \rho\widehat{f}, \nabla\ln\frac{\rho}{\nu}\right\rangle$$

$$= \left\langle\rho\left(f - \widehat{f}\right), \nabla\ln\frac{\rho}{\nu}\right\rangle + \left\langle\frac{\rho}{\nu}\nabla\cdot(\nu\widehat{J}) - \nabla\cdot(\rho J), \nabla\ln\frac{\rho}{\nu}\right\rangle \quad (33)$$

where the first equality is by $\nabla\frac{\rho}{\nu} = \frac{\rho}{\nu}\nabla\ln\frac{\rho}{\nu}$. The derivation above splits $\frac{d}{dt}\mathrm{KL}$ into two terms: one depends on the drifts $\widehat{f}$ and $f$, while the other depends on the diffusion terms $\widehat{J}$ and $J$. Observe that the drift-related term matches the second term in the final result, which vanishes when $\widehat{f} = f$.

We now focus on the remaining, diffusion-related term:

$$\left\langle\frac{\rho}{\nu}\nabla\cdot\left(\nu\widehat{J}\right) - \nabla\cdot(\rho J), \nabla\ln\frac{\rho}{\nu}\right\rangle$$

$$= \left\langle\frac{\rho}{\nu}\left(\widehat{J}\nabla\nu + \nu\nabla\cdot\widehat{J}\right) - J\nabla\rho - \rho\nabla\cdot J, \nabla\ln\frac{\rho}{\nu}\right\rangle$$

$$= \rho\left\langle\widehat{J}\nabla\ln\nu + \nabla\cdot\widehat{J} - J\nabla\ln\rho - \nabla\cdot J, \nabla\ln\frac{\rho}{\nu}\right\rangle$$

$$= \rho\left\langle-\widehat{J}\nabla\ln\frac{\rho}{\nu} + (\widehat{J} - J)\nabla\ln\rho + \nabla\cdot(\widehat{J} - J), \nabla\ln\frac{\rho}{\nu}\right\rangle$$

$$= -\rho\left\|\nabla\ln\frac{\rho}{\nu}\right\|_{\widehat{J}}^2 + \rho\left\langle(\widehat{J} - J)\nabla\ln\rho + \nabla\cdot(\widehat{J} - J), \nabla\ln\frac{\rho}{\nu}\right\rangle. \quad (34)$$

Here, the first equality is by $\nabla\cdot\left(\nu\widehat{J}\right) = \widehat{J}\nabla\nu + \nu\nabla\cdot\widehat{J}$ and $\nabla\cdot(\rho J) = J\nabla\rho + \rho\nabla\cdot J$, and the second equality uses $\frac{1}{\nu}\nabla\nu = \nabla\ln\nu$ and $\frac{1}{\rho}\nabla\rho = \nabla\ln\rho$. Combining Eqs. (32) to (34) yields the desired result. $\qquad\square$

# D  Proofs of auxiliary lemmata

## D.1  Proof of Lemma 1 – Interpolation SDE for the Backward Algorithm

For ease of notation, within this proof we omit the step index $k$ in $\widehat{X}_t^{(k)}$, $\widehat{\mu}_k$, and $\widehat{G}_k$. For $t \geq 0$, let

$$\tilde{X}_t = (1 - t)\widehat{X}_t - 2t\nabla\ln p_{T-kh-t}\left(\widehat{X}_t\right)$$

so $\tilde{X}_0 = \widehat{X}_0$, and we can write (18) as

$$\tilde{X}_t = \tilde{X}_0 + \sqrt{2}W_t.$$

That is, $\tilde{X}_t$ evolves following the SDE

$$d\tilde{X}_t = \sqrt{2}dW_t. \quad (35)$$

Suppose that the process $(\widehat{X}_t)_{t\in[0,h]}$ evolves via an SDE of the form

$$d\widehat{X}_t = \widehat{\mu}\left(\widehat{X}_t, t\right)dt + \sqrt{2\widehat{G}\left(\widehat{X}_t, t\right)}dW_t$$

for some functions $\widehat{\mu}$ and $\widehat{G}$. In what follows, we solve for the form of these two functions. Let $T_t(x) = (1-t)\,x - 2t\nabla \ln p_{T-kh-t}(x) \in \mathbb{R}^d$. Then, we have

$$\frac{\partial T_t}{\partial t}(x) = -x - 2\nabla \ln p_{T-kh-t}(x) - 2t\frac{\partial}{\partial t}\nabla \ln p_{T-kh-t}(x),$$

$$\nabla T_t(x) = (1-t)\,I_d - 2t\nabla^2 \ln p_{T-kh-t}(x), \quad \text{and}$$

$$\nabla^2 T_t(x) = -2t\nabla^3 \ln p_{T-kh-t}(x).$$

We now apply high-dimensional Itô's lemma [86, Chapter 4.2] to $\tilde{X}_t = T_t\left(\widehat{X}_t\right)$ and obtain the following

$$
\begin{aligned}
d\tilde{X}_t &= dT_t\left(\widehat{X}_t\right)\\[4pt]
&= \frac{\partial T_t}{\partial t}\left(\widehat{X}_t\right)dt + \nabla T_t\left(\widehat{X}_t\right)d\widehat{X}_t + \sum_{i,j=1}^d \frac{1}{2}\frac{\partial^2 T_t\left(\widehat{X}_t\right)}{\partial x_i \partial x_j}\,d\widehat{X}_t^i\,d\widehat{X}_t^j\\[4pt]
&= \frac{\partial T_t}{\partial t}\left(\widehat{X}_t\right)dt + \nabla T_t\left(\widehat{X}_t\right)\widehat{\mu}dt + \nabla T_t\left(\widehat{X}_t\right)\sqrt{2\widehat{G}}\,dW_t + \sum_{i,j=1}^d \frac{\partial^2 T_t\left(\widehat{X}_t\right)}{\partial x_i \partial x_j}\,\widehat{G}_{i,j}dt\\[4pt]
&= \left(\frac{\partial T_t}{\partial t}\left(\widehat{X}_t\right) + \nabla T_t(\widehat{X}_t)\widehat{\mu} + \mathrm{Tr}\left[\nabla^2 T_t\left(\widehat{X}_t\right)\widehat{G}\right]\right)dt + \sqrt{2}\nabla T_t\left(\widehat{X}_t\right)\sqrt{\widehat{G}}\,dW_t. \quad (36)
\end{aligned}
$$

where the third equality is by plugging in $d\widehat{X}_t = \widehat{\mu}dt + \sqrt{2\widehat{G}}dW_t$ and noting that

$$d\widehat{X}_t^i d\widehat{X}_t^j = 2\widehat{G}_{i,j}dt,$$

and the last step uses $\mathrm{Tr}(AB^T) = \sum_{i,j} A_{i,j}B_{i,j}$. Note that in the equations above, $T_t(\cdot) \in \mathbb{R}^d$ is a vector and therefore, $\frac{\partial^2 T_t(\cdot)}{\partial x_i \partial x_j} \in \mathbb{R}^d$, $\nabla^2 T_t(\cdot) \in \mathbb{R}^{d\times d\times d}$, and $\mathrm{Tr}[\nabla^2 T_t(\cdot)G] \in \mathbb{R}^d$ are obtained by broadcasting the corresponding operator to each entry in $T_t(\cdot)$.

Matching (35) and (36), we must have

$$\frac{\partial T_t}{\partial t}(x) + \nabla T_t(x)\widehat{\mu} + \mathrm{Tr}\left[\nabla^2 T_t(x)\widehat{G}\right] = 0,$$

$$\nabla T_t(x)\sqrt{\widehat{G}} = I_d$$

which implies

$$\widehat{G}(x,t) = [\nabla T_t(x)]^{-2}$$

$$\widehat{\mu}(x,t) = \sqrt{\widehat{G}}\left\{-\frac{\partial T_t}{\partial t}(x) - \mathrm{Tr}\left[\nabla^2 T_t(x)\widehat{G}\right]\right\}.$$

Plugging in $\frac{\partial T_t}{\partial t}(x)$, $\nabla T_t(x)$, and $\nabla^2 T_t(x)$ completes the proof. $\qquad\square$

### D.2 Proof of Lemma 2 – Interpolation SDE for the Hybrid Algorithm

Observe that the next iterate of (16), $\widehat{X}_{(k+1)h}$, is the value at time $t = h$ of the stochastic process $(\widehat{X}_t^{(k)})_{t\in[0,h]}$ given by

$$\widehat{X}_t^{(k)} = \widehat{X}_0^{(k)} + t\widehat{X}_0^{(k)} + 2t\nabla \ln p_{T-kh-t}(\widehat{X}_t^{(k)}) + \sqrt{2}W_t, \quad \widehat{X}_0^{(k)} = \widehat{X}_{kh}. \quad (37)$$

For $t \ge 0$, let

$$\tilde{X}_t = \widehat{X}_t^{(k)} - t\widehat{X}_0^{(k)} - 2t\nabla \ln p_{T-kh-t}(\widehat{X}_t^{(k)})$$

so $\tilde{X}_0 = \widehat{X}_0^{(k)}$ and $\tilde{X}_t^{(k)} = \tilde{X}_0^{(k)} + \sqrt{2}W_t$.

The rest of the argument is analogous to the proof of Lemma 1 (see Appendix D.2), but with $T_t(x) = x - t\widehat{X}_0^{(k)} - 2t\nabla \ln p_{T-kh-t}(x) \in \mathbb{R}^d$. $\qquad\square$

### D.3 Proof of Lemma 3 – Second Moment of OU Process

This is a standard property of the OU process. We include a proof here for completeness. Along the OU process (12), we have

$$X_t \stackrel{d}{=} e^{-t}X_0 + \sqrt{1 - e^{-2t}}Z, \quad Z \sim \mathcal{N}(0, I).$$

Therefore,

$$
\begin{aligned}
\mathbb{E}\left[\|X_t\|^2\right] &= \mathbb{E}\left[\left\|e^{-t}X_0 + \sqrt{1 - e^{-2t}}Z\right\|^2\right] \\
&\leq 2e^{-2t}\mathbb{E}\left[\|X_0\|^2\right] + 2(1 - e^{-2t})\mathbb{E}\left[\|Z\|^2\right] \\
&= 2e^{-2t}\mathbb{E}\left[\|X_0\|^2\right] + 2(1 - e^{-2t})d \\
&\leq 2\mathbb{E}\left[\|X_0\|^2\right] + 2d
\end{aligned}
$$

where the second line follows by Young's inequality, completing the proof of Lemma 3. □

### D.4 Proof of Lemma 4 – Bound on Initialization Mismatch

*Proof.* Recall that $p_t$ denotes the density of the OU process at time $t$, thus $p_T = \overleftarrow{p}_0$. We have

$$
\begin{aligned}
\mathrm{KL}(p_t\|q_0) &= \int_{\mathbb{R}^d} p_t(x)\ln p_t(x)dx - \int_{\mathbb{R}^d} p_t(x)\ln q_0(x)dx \\
&= \int_{\mathbb{R}^d} p_t(x)\ln p_t(x)dx + \int_{\mathbb{R}^d} p_t(x)\left(\frac{\|x\|^2}{2} + \frac{d}{2}\ln(2\pi)\right)dx \\
&\leq \int_{\mathbb{R}^d} p_t(x)\ln p_t(x)dx + \frac{1}{2}\mathbb{E}_{p_t}[\|x\|^2] + \frac{d}{2}\ln(2\pi).
\end{aligned}
$$

For the first term, letting $p_{t|0}$ be the conditional density of $X_t$ given $X_0$, we have

$$
\begin{aligned}
\int_{\mathbb{R}^d} p_t(x)\ln p_t(x)\,dx &= \int_{\mathbb{R}^d}\left(\int_{\mathbb{R}^d} p_{t|0}(x|y)p_0(y)dy\right)\ln\left(\int_{\mathbb{R}^d} p_{t|0}(x|y)\,p_0(y)dy\right)dx \\
&\leq \int_{\mathbb{R}^d}\left[\int_{\mathbb{R}^d} p_{t|0}(x|y)\ln p_{t|0}(x|y)\,p_0(y)dy\right]dx \\
&= \int_{\mathbb{R}^d}\left(\int_{\mathbb{R}^d} p_{t|0}(x|y)\ln p_{t|0}(x|y)\,dx\right)p_0(y)dy
\end{aligned}
$$

where the inequality is by Jensen's inequality and that $x \mapsto x\ln x$ is a convex function for $x > 0$. Since $X_t \mid X_0 \sim \mathcal{N}(e^{-t}X_0, (1 - e^{-2t})I)$, by entropy of Gaussian distributions, we have

$$\int_{\mathbb{R}^d} p_{t|0}(x|y)\ln p_{t|0}(x|y)\,\mathrm{d}x = -\frac{d}{2}\ln(2\pi(1 - e^{-2t})) - \frac{d}{2}.$$

Therefore,

$$
\begin{aligned}
\mathrm{KL}(p_t\|q_0) &\leq -\frac{d}{2}\ln(2\pi(1 - e^{-2t})) - \frac{d}{2} + \frac{1}{2}\mathbb{E}_{p_t}[\|x\|^2] + \frac{d}{2}\ln(2\pi) \\
&= \frac{1}{2}\mathbb{E}_{p_t}[\|x\|^2] + \frac{d}{2}\left(\ln\frac{1}{1 - e^{-2t}} - 1\right) \\
&\leq M_2 + d + \frac{d}{2}\left(\ln\frac{1}{1 - e^{-2t}} - 1\right)
\end{aligned}
$$

where the last inequality is by Lemma 3. Taking $t' = \frac{1}{2}\ln(\frac{e}{e-1}) \approx 0.229$, we have $\ln\frac{1}{1-e^{-2t'}} = 1$, thus

$$\mathrm{KL}(p_{t'}\|q_0) \leq M_2 + d.$$

By exponential convergence of the OU process, we have for all $T \geq t'$,

$$\mathrm{KL}(p_T\|q_0) \leq e^{-2(T-t')}\mathrm{KL}(p_{t'}\|q_0) \leq e^{-2(T-t')} \cdot (M_2 + d) < 2e^{-2T}(M_2 + d)$$

as desired.

□

## D.5 Proof of Lemma 6 – Bounds on the Diffusion Matrix $\widehat{G}_k$ for Backward Algorithm

Since $p_{T-kh-t}$ is $L$-smooth,

$$-LI \preceq \nabla^2 \ln p_{T-kh-t}(x) \preceq LI.$$

So

$$(1-t-2tL)I \preceq (1-t)I_d - 2t\nabla^2 \ln p_{T-kh-t}(x) \preceq (1-t+2tL)I.$$

Check that $0 < 1-t-2tL \le 1-t+2tL$ since $t \le \frac{1}{2(2L+1)}$. Therefore,

$$(1-t+2tL)^{-2}I \preceq \widehat{G}_k = [(1-t)I_d - 2t\nabla^2 \ln p_{T-kh-t}(x)]^{-2} \preceq (1-t-2tL)^{-2}I$$

and

$$(1-t+2tL)^{-1}I \preceq \sqrt{\widehat{G}_k} = [(1-t)I_d - 2t\nabla^2 \ln p_{T-kh-t}(x)]^{-1} \preceq (1-t-2tL)^{-1}I.$$

We collect some useful facts:

$$(1-x)^{-2} \le 1+6x \quad \text{if } 0 \le x \le \frac{1}{2},$$
$$(1-x)^{-2} \ge 1+2x \quad \text{if } x < 1,$$
$$(1-x)^{-1} \le 1+2x \quad \text{if } 0 \le x \le \frac{1}{2},$$
$$(1-x)^{-1} \ge 1+x \quad \text{if } x < 1.$$

Since $t - 2tL \le \frac{1}{2}$, $0 \le t + 2tL \le \frac{1}{2}$, and $L \ge 1$,

$$(1-4tL)I \preceq [1+2t(1-2L)]I \preceq \widehat{G}_k \preceq [1+6t(1+2L)]I \preceq (1+18tL)I$$

and

$$(1-2tL)I \preceq [1+t(1-2L)]I \preceq \sqrt{\widehat{G}_k} \preceq [1+2t(1+2L)]I \preceq (1+6tL)I$$

as desired.

## D.6 Proof of Lemma 9 – Bounds on the Diffusion Matrix $\widehat{G}_k$ for Hybrid Algorithm

The proof is analogous to that of Lemma 6 (see Appendix D.5). The resulting bound is identical up to changes in constant factors.

## D.7 Proof of Lemma 7 – Bound on the Drift Difference $\mu_k - \widehat{\mu}_k$ for Backward Algorithm

We start by establishing a few auxiliary lemmas (Lemmas 14 to 16). We let $[b_i]_{i=1}^d \in \mathbb{R}^d$ denote a vector whose $i$-th entry is $b_i$.

**Lemma 14.** *The following two equalities hold true.*

$$\frac{\partial}{\partial t} \ln p_t = \|\nabla \ln p_t\|^2 + \langle \nabla \ln p_t, x \rangle + \Delta(\ln p_t) + d, \quad \text{and}$$

$$\frac{\partial}{\partial t} \nabla \ln p_{T-kh-t}(x) = -\text{Tr}(\nabla^3 \ln p_{T-kh-t}(x)\widehat{G}_k) - \nabla \left( \|\nabla \ln p_{T-kh-t}\|^2 \right)$$
$$- \nabla \left( \langle \nabla \ln p_{T-kh-t}, x \rangle \right) + \left[ \text{Tr}(\nabla_i \nabla^2 \ln p_{T-kh-t}(x)(\widehat{G}_k - I)) \right]_{i=1}^d.$$

*Proof.* Applying the Fokker-Planck equation to (12) yields $\frac{\partial p_t}{\partial t} = \nabla \cdot [p_t (\nabla \ln p_t + x)]$. Therefore,

$$\frac{\partial}{\partial t} \ln p_t = \frac{1}{p_t} \frac{\partial p_t}{\partial t}$$
$$= \frac{1}{p_t} \nabla \cdot [p_t (\nabla \ln p_t + x)]$$
$$= \frac{1}{p_t} \left( \langle \nabla p_t, \nabla \ln p_t + x \rangle + p_t \nabla \cdot (\nabla \ln p_t + x) \right)$$
$$= \langle \nabla \ln p_t, \nabla \ln p_t \rangle + \langle \nabla \ln p_t, x \rangle + \Delta(\ln p_t) + d$$

where the second equality uses $\nabla \cdot (fG) = \langle \nabla f, G \rangle + f \nabla \cdot G$, for $f\colon \mathbb{R}^d \to \mathbb{R}$ and $G\colon \mathbb{R}^d \to \mathbb{R}^d$. We have shown the first result.

Then, note that

$$\frac{\partial}{\partial t} \ln p_{T-kh-t}(x) = -\frac{\partial}{\partial t'} \ln p_{t'}(x)\bigg|_{t'=T-kh-t}$$

$$= -\|\nabla \ln p_{T-kh-t}\|^2 - \langle \nabla \ln p_{T-kh-t}, x \rangle - \Delta(\ln p_{T-kh-t}) - d.$$

Therefore,

$$\nabla\left(\frac{\partial}{\partial t} \ln p_{T-kh-t}(x)\right) = -\nabla\left(\|\nabla \ln p_{T-kh-t}\|^2\right) - \nabla\left(\langle \nabla \ln p_{T-kh-t}, x \rangle\right) - \nabla\left(\Delta(\ln p_{T-kh-t})\right)$$

$$= -\nabla\left(\|\nabla \ln p_{T-kh-t}\|^2\right) - \nabla\left(\langle \nabla \ln p_{T-kh-t}, x \rangle\right) - \left[\mathrm{Tr}(\nabla_i \nabla^2 \ln p_{T-kh-t})\right]_{i=1}^d.$$

Recall that

$$\mathrm{Tr}(\nabla^3 \ln p_{T-kh-t}(x)\widehat{G}_k) = \left[\mathrm{Tr}(\nabla_i \nabla^2 \ln p_{T-kh-t}(x)\widehat{G}_k)\right]_{i=1}^d.$$

Adding the two equations above yields the second result. □

**Lemma 15.** *The following inequality holds.*

$$\left\|\left[\mathrm{Tr}(\nabla_i \nabla^2 \ln p_{T-kh-t}(x)(\widehat{G}_k - I))\right]_{i=1}^d\right\|^2 \le 324t^2 d^3 L^2 H_{T-kh-t}(x)^2.$$

*Proof.* For each entry, we have

$$\left|\mathrm{Tr}(\nabla_i \nabla^2 \ln p_{T-kh-t}(x)(\widehat{G}_k - I))\right|^2 \le \|\nabla_i \nabla^2 \ln p_{T-kh-t}(x)\|_\mathrm{F}^2 \|\widehat{G}_k - I\|_\mathrm{F}^2$$

$$\le d^2 \left\|\nabla_i \nabla^2 \ln p_{T-kh-t}(x)\right\|_\mathrm{op}^2 \left\|\widehat{G}_k - I\right\|_\mathrm{op}^2$$

$$\le 324t^2 d^2 L^2 H_{T-kh-t}(x)^2.$$

Summing over the entries yields the desired result. □

**Lemma 16.** *The following inequality holds.*

$$\left\|\frac{\partial}{\partial t}\nabla \ln p_{T-kh-t}(x) + \mathrm{Tr}(\nabla^3 \ln p_{T-kh-t}(x)\widehat{G}_k)\right\| \le (2L+1)\|\nabla \ln p_{T-kh-t}\| + L\|x\| + 18td^{\frac{3}{2}}LH_{T-kh-t}(x).$$

*Proof.* By Lemma 14, we have

$$\left\|\frac{\partial}{\partial t}\nabla \ln p_{T-kh-t}(x) + \mathrm{Tr}(\nabla^3 \ln p_{T-kh-t}(x)\widehat{G}_k)\right\|$$

$$\le \left\|\nabla\left(\|\nabla \ln p_{T-kh-t}\|^2\right)\right\| + \left\|\nabla\left(\langle \nabla \ln p_{T-kh-t}, x\rangle\right)\right\| + \left\|\left[\mathrm{Tr}(\nabla_i \nabla^2 \ln p_{T-kh-t}(x)(\widehat{G}_k - I))\right]_{i=1}^d\right\|.$$

From this point onward, we let $\nu_t = p_{T-kh-t}$ for ease of notation. Next, we bound the three terms in this sum For the first term, we have

$$\nabla\left(\|\nabla \ln \nu_t\|^2\right) = \nabla\left(\langle \nabla \ln \nu_t, \nabla \ln \nu_t\rangle\right) = 2\nabla^2 \ln \nu_t \nabla \ln \nu_t$$

where the second equality uses $\nabla(\langle f, g\rangle) = J_f^T g + J_g^T f$, where $f, g\colon \mathbb{R}^n \to \mathbb{R}^m$, and $J_f$ is the Jacobian of $f$. Therefore,

$$\left\|\nabla\left(\|\nabla \ln \nu_t\|^2\right)\right\| \le 2\left\|\nabla^2 \ln \nu_t\right\|_\mathrm{op} \|\nabla \ln \nu_t\| \le 2L\|\nabla \ln \nu_t\|.$$

To bound the second term, note that $\nabla(\langle \nabla \ln \nu_t, x\rangle) = \nabla \ln \nu_t + \nabla^2 \ln \nu_t x$. So

$$\|\nabla(\langle \nabla \ln \nu_t, x\rangle)\| \le \|\nabla \ln \nu_t\| + \left\|\nabla^2 \ln \nu_t\right\|_\mathrm{op} \|x\| \le \|\nabla \ln \nu_t\| + L\|x\|.$$

Lemma 15 gives a bound on the third term. Combining the three upper bounds finishes the proof of Lemma 16. □

We are ready to prove Lemma 7. We start by decomposing $\widehat{\mu}_k - \mu_k$ as

$$\widehat{\mu}_k - \mu_k = \left( \sqrt{\widehat{G}_k} - I \right) (x + 2\nabla \ln p_{T-kh-t}(x))$$
$$+ 2t\sqrt{\widehat{G}_k} \left( \frac{\partial}{\partial t} \nabla \ln p_{T-kh-t}(x) + \mathrm{Tr}\left( \nabla^3 \ln p_{T-kh-t}(x)\widehat{G}_k \right) \right).$$

Next, we bound the two terms on the right-hand side. For the first term, we have

$$\left\| \left( \sqrt{\widehat{G}_k} - I \right) (x + 2\nabla \ln p_{T-kh-t}(x)) \right\| \leq \left\| \sqrt{\widehat{G}_k} - I \right\|_{\mathrm{op}} \|x + 2\nabla \ln p_{T-kh-t}(x)\|$$
$$\leq 6tL \left( \|x\| + 2\|\nabla \ln p_{T-kh-t}(x)\| \right)$$

where the second inequality uses the bound $\left\| \sqrt{G} - I \right\|_{\mathrm{op}} \leq 6tL$ by Lemma 6.

For the second term, we have

$$\left\| 2t\sqrt{\widehat{G}_k} \left( \frac{\partial}{\partial t} \nabla \ln p_{T-kh-t}(x) + \mathrm{Tr}\left( \nabla^3 \ln p_{T-kh-t}(x)\widehat{G}_k \right) \right) \right\|$$
$$\leq 2t \left\| \sqrt{\widehat{G}_k} \right\|_{\mathrm{op}} \left\| \frac{\partial}{\partial t} \nabla \ln p_{T-kh-t}(x) + \mathrm{Tr}\left( \nabla^3 \ln p_{T-kh-t}(x)\widehat{G}_k \right) \right\|$$
$$\leq 5t \left[ (2L+1) \|\nabla \ln p_{T-kh-t}\| + L\|x\| + 18td^{\frac{3}{2}} L H_{T-kh-t}(x) \right].$$

where the second inequality is by Lemma 16 and $\left\| \sqrt{\widehat{G}_k} \right\|_{\mathrm{op}} \leq \frac{5}{2}$ from Lemma 6.

The desired bound follows from combining the above, noting that $L \geq 1$ and using $(a + b + c)^2 \leq 3a^2 + 3b^2 + 3c^2$ for $a, b, c \in \mathbb{R}$. $\qquad\square$

## D.8 Proof of Lemma 10 – Bound on the Drift Difference $\mu_k - \widehat{\mu}_k$ for Hybrid Algorithm

The proof follows the same structure as that of Lemma 7 (see Appendix D.7). We start by decomposing $\widehat{\mu}_k - \mu_k$ as

$$\widehat{\mu}_k - \mu_k = (\sqrt{\widehat{G}_k} - I)(x + 2\nabla \ln p_{T-kh-t}(x))$$
$$+ 2t\sqrt{\widehat{G}_k} \left( \frac{\partial}{\partial t} \nabla \ln p_{T-kh-t}(x) + \mathrm{Tr}\left( \nabla^3 \ln p_{T-kh-t}(x)\widehat{G}_k \right) \right)$$
$$+ \sqrt{\widehat{G}_k} (a - x) .$$

The first two terms are bounded analogously to the proof of Lemma 7 (see Appendix D.7). The last term is bounded by $\left\| \sqrt{\widehat{G}_k} \right\|_{\mathrm{op}} \leq 2$ thanks to Lemma 9. This completes the proof. $\qquad\square$

## D.9 Proof of Lemma 8 – Bound on the Divergence of Diffusion Matrix $\nabla \cdot \widehat{G}_k$ for Backward Algorithm

In this section, we prove Lemma 8, which provides a bound on $\|\nabla \cdot \widehat{G}_k\|$. The derivation in this section is largely inspired by [79, Proof of Lemma 6], but we improved the technique to obtain a tighter bound in terms of $d$ and with weaker assumptions.

Before proving Lemma 8, we show a few general lemmas related to matrix perturbation (Lemmas 17 and 18) and divergence of matrix-valued functions (Lemma 19). These results are then used to prove Lemma 8. The next two lemmas are folklore; we include their proofs for completeness. Recall that a matrix norm $\| \cdot \|$ is sub-multiplicative if for all $A, B$ we have

$$\|AB\| \leq \|A\|\|B\|.$$

**Lemma 17.** *Let $A$ and $B$ be real invertible matrices. Let $\|\cdot\|$ be a sub-multiplicative matrix norm. Then, the following three inequalities hold true*

$$\left\|A^2 - B^2\right\| \leq 2\left\|A - B\right\|(\|A\| + \|B\|),$$
$$\left\|A^{-1} - B^{-1}\right\| \leq \|A - B\|\left\|A^{-1}\right\|\left\|B^{-1}\right\|, \quad \text{and}$$
$$\left\|A^{-2} - B^{-2}\right\| \leq 2\|A - B\|(\|A\| + \|B\|)\left\|A^{-2}\right\|\left\|B^{-2}\right\|.$$

*Proof.* We start with the first inequality. By the triangular inequality and sub-multiplicativity, we have

$$\left\|A^2 - B^2\right\| = \left\|(B + A - B)^2 - B^2\right\|$$
$$= \left\|(A - B)^2 + B(A - B) + (A - B)B\right\|$$
$$\leq \|A - B\|^2 + 2\|A - B\|\|B\|.$$

A combination of symmetry and the triangle inequality yields

$$\left\|A^2 - B^2\right\| \leq \|A - B\|^2 + 2\|A - B\|\min(\|A\|, \|B\|)$$
$$\leq \|A - B\|(\|A\| + \|B\| + 2\min(\|A\|, \|B\|))$$
$$\leq 2\|A - B\|(\|A\| + \|B\|)$$

as desired.

The second inequality follows from the identity $A^{-1} - B^{-1} = A^{-1}(B - A)B^{-1}$ and the sub-multiplicativity of the operator norm.

Iteratively applying the first and second results gives the final inequality

$$\left\|A^{-2} - B^{-2}\right\| \leq \left\|A^2 - B^2\right\|\left\|A^{-2}\right\|\left\|B^{-2}\right\| \leq 2\|A - B\|(\|A\| + \|B\|)\left\|A^{-2}\right\|\left\|B^{-2}\right\|,$$

this concludes the proof of Lemma 17. $\qquad\square$

**Lemma 18.** *Let $A\colon \mathbb{R}^n \to \mathbb{R}^{m\times m}$ be a matrix-valued function. Assume at a point $x$, $A$ is continuous and differentiable, and $[A(x)]^{-2}$ exists within a neighborhood of $x$. Let $\|\cdot\|$ be a sub-multiplicative matrix norm. Let $\varphi(x) = A(x)^{-2}$ and fix a norm-one vector $v \in \mathbb{R}^n$, then*

$$\|\nabla_v\varphi(x)\| \leq 4\|A(x)\|\left\|[A(x)]^{-2}\right\|^2\|\nabla_v A(x)\|.$$

*Proof.* Let $\varphi(x) = [\varphi(x)]^{-2}$. We have

$$\|\nabla_v\varphi(x)\| = \left\|\lim_{u\to 0}\frac{\varphi(x + uv) - \varphi(x)}{u}\right\|$$
$$= \lim_{u\to 0}\frac{\|\varphi(x + uv) - \varphi(x)\|}{u}$$
$$\leq \lim_{u\to 0}\frac{2\|A(x + uv) - A(x)\|(\|A(x + uv)\| + \|A(x)\|)\|\varphi(x + uv)\|\|\varphi(x)\|}{u}$$
$$= 4\|A(x)\|\|\varphi(x)\|^2\lim_{u\to 0}\frac{\|A(x + uv) - A(x)\|}{u}$$
$$= 4\|A(x)\|\|\varphi(x)\|^2\|\nabla_v A(x)\|$$

where the inequality is by Lemma 17. $\qquad\square$

Next, we establish a general result that bounds the divergence of a matrix-valued function by the norm of its derivatives.

**Lemma 19.** *Let $M\colon \mathbb{R}^d \to \mathbb{R}^{d\times d}$ be a matrix-valued function. (Note that $M(x)$ does not need to be a symmetric matrix.) Then, $\|\nabla \cdot M(x)\|^2 \leq d\sum_{j=1}^d \|\nabla_j M(x)\|_{\text{op}}^2$.*

*Proof.* Denote the $i, j$-th entry of a matrix $A$ by $A_{ij}$. Recall that the $\ell_2$ norm of each row or column vector of a matrix is upper bounded by the operator norm of the matrix. Thus,

$$\sum_{i=1}^{d} \left( \frac{\partial M_{ij}(x)}{\partial x_j} \right)^2 = \sum_{i=1}^{d} (\nabla_j M(x))_{ij}^2 \leq \|\nabla_j M(x)\|_{\mathrm{op}}^2 .$$

Hence, applying $\|u\|_1^2 \leq d\|u\|_2^2$ for all $u \in \mathbb{R}^d$ yields

$$\|\nabla \cdot M(x)\|^2 = \sum_{i=1}^{d} \left( \sum_{j=1}^{d} \frac{\partial M_{ij}(x)}{\partial x_j} \right)^2 \leq d \sum_{i=1}^{d} \sum_{j=1}^{d} \left( \frac{\partial M_{ij}(x)}{\partial x_j} \right)^2 \leq d \sum_{j=1}^{d} \|\nabla_j M(x)\|_{\mathrm{op}}^2,$$

as desired. This concludes the proof of Lemma 19. $\qquad\square$

Finally, we prove Lemma 8.

*Proof of Lemma 8.* By Lemma 19, we have $\|\nabla \cdot \widehat{G}_k\|^2 \leq d \sum_{j=1}^{d} \left\| \nabla_j \widehat{G}_k \right\|_{\mathrm{op}}^2$. We now claim that for each $j = 1, \ldots, d$,

$$\left\| \nabla_j \widehat{G}_k \right\|_{\mathrm{op}} \leq 364\, t H_{T-kh-t}(x)$$

which will imply the desired bound $\|\nabla \cdot \widehat{G}_k\| \leq 364\, t d H_{T-kh-t}(x)$. Let $T(x) = (1-t)I_d - 2t\nabla^2 \ln p_{T-kh-t}(x)$ so that $\widehat{G}_k(x) = [T(x)]^{-2}$. We invoke Lemma 18 with $v = e_j$ (note that $\|\cdot\|_{\mathrm{op}}$ is sub-multiplicative) to obtain

$$\left\| \nabla_j \widehat{G}_k(x) \right\|_{\mathrm{op}} \leq 4 \|T(x)\|_{\mathrm{op}} \left\| \widehat{G}_k(x) \right\|_{\mathrm{op}}^2 \|\nabla_j T(x)\|_{\mathrm{op}} .$$

Note that both $\|T(x)\|_{\mathrm{op}}$ and $\|\widehat{G}_k(x)\|_{\mathrm{op}}$ are bounded by constants. On the one hand, Lemma 6 ensures $\|\widehat{G}_k\|_{\mathrm{op}} \leq \frac{11}{2}$. On the other, since $\left\| \nabla^2 \ln p_{T-kh-t} \right\|_{\mathrm{op}} \leq L$ we have

$$\|T(x)\|_{\mathrm{op}} \leq 1 - t + 2tL \leq 1 + 2tL < \frac{3}{2}$$

where the last inequality follows since $t \leq \frac{1}{2(2L+1)}$ and so $tL < 1/4$. Plugging in the bounds for $\|T(x)\|_{\mathrm{op}}$ and $\|\widehat{G}_k(x)\|_{\mathrm{op}}$ gives

$$\left\| \nabla_j \widehat{G}_k(x) \right\|_{\mathrm{op}} \leq 182 \|\nabla_j T(x)\|_{\mathrm{op}} = 364\, t \left\| \nabla_j \nabla^2 \ln p_{T-kh-t}(x) \right\|_{\mathrm{op}} \leq 364\, t H_{T-kh-t}(x)$$

as desired; concluding the proof Lemma 8. $\qquad\square$

## D.10 Proof of Lemma 11 – Bound on the Divergence of Diffusion Matrix $\nabla \cdot \widehat{G}_k$ for Hybrid Algorithm

The proof follows the same structure as that of Lemma 8 (see Appendix D.9). The key difference is that we let

$$T(x) = I_d - 2t\nabla^2 \ln p_{T-kh-t}(x)$$

so that $\widehat{G}_k(x) = [T(x)]^{-2}$. The resulting bound only differs in the constant factor.

## D.11 Proof of Lemma 13

For ease of notation, we let $\nu_0 = \overleftarrow{p}_0^{(k)}$, $\nu_t = \overleftarrow{p}_t^{(k)}$, $\nu_{t|0} = \overleftarrow{p}_{t|0}^{(k)}$ and $\nu_{0|t} = \overleftarrow{p}_{0|t}^{(k)}$ within this proof.

By Bayes rule, we have

$$\nabla_x \ln \nu_{t|0}(x \mid a) = \nabla_x \left[ \ln \nu_{0|t}(a \mid x) + \ln \nu_t(x) \right] .$$

Let $g(z; \mu, \Sigma)$ denote the density of $\mathcal{N}(\mu, \Sigma)$ at $z$. Then,

$$
\nu_{0|t}(a \mid x) = p_{T-kh|T-kh-t}(a \mid x)
$$
$$
= g(a; e^{-t}x, (1 - e^{-2t})I).
$$

Therefore,

$$
\nabla_x \ln \nu_{0|t}(a \mid x) = \nabla_x \left[ -\frac{\|a - e^{-t}x\|^2}{2(1 - e^{-2t})} \right]
$$
$$
= \frac{e^{-t}(a - e^{-t}x)}{1 - e^{-2t}}.
$$

Plugging the above into the Bayes rule identity, we have

$$
\nabla_x \ln \nu_{t|0}(x \mid a) = \frac{e^{-t}(a - e^{-t}x)}{1 - e^{-2t}} + \nabla_x \ln \nu_t(x).
$$

Therefore,

$$
\mathbb{E}_{(a,x) \sim (\nu_0, \nu_t)} \left[ \|\nabla_x \ln \nu_{t|0}(x \mid a)\|^2 \right] = \mathbb{E}_{(a,x) \sim (\nu_0, \nu_t)} \left[ \left\| \frac{e^{-t}(a - e^{-t}x)}{1 - e^{-2t}} + \nabla_x \ln \nu_t(x) \right\|^2 \right]
$$
$$
\lesssim \mathbb{E}_{(a,x) \sim (\nu_0, \nu_t)} \left[ \left\| \frac{e^{-t}(a - e^{-t}x)}{1 - e^{-2t}} \right\|^2 + \|\nabla_x \ln \nu_t(x)\|^2 \right]
$$
$$
= \frac{e^{-2t}}{(1 - e^{-2t})^2} \mathbb{E}_{(a,x) \sim (\nu_0, \nu_t)} \left[ \|a - e^{-t}x\|^2 \right] + \mathbb{E}_{\nu_t} \left[ \|\nabla_x \ln \nu_t(x)\|^2 \right].
$$

Then, since $\nu_{0|t}(\cdot \mid x) = \mathcal{N}(e^{-t}x, (1 - e^{-2t})I)$, we have

$$
\mathbb{E}_{(a,x) \sim (\nu_0, \nu_t)} \left[ \|a - e^{-t}x\|^2 \right] = \mathbb{E}_{x \sim \nu_t} \mathbb{E}_{a \sim \nu_{0|t}(\cdot|x)} \left[ \|a - e^{-t}x\|^2 \right]
$$
$$
= d(1 - e^{-2t}).
$$

Therefore,

$$
t^2 L^2 \, \mathbb{E}_{(a,x) \sim (\nu_0, \nu_t)} \left[ \|\nabla \ln \nu_{t|0}(x \mid a)\|^2 \right] \lesssim t^2 L^2 \left\{ \frac{d e^{-2t}}{1 - e^{-2t}} + \mathbb{E}_{\nu_t} \left[ \|\nabla_x \ln \nu_t(x)\|^2 \right] \right\}
$$
$$
= dL^2 \frac{t^2 e^{-2t}}{1 - e^{-2t}} + t^2 L^2 \mathbb{E}_{\nu_t} \left[ \|\nabla_x \ln \nu_t(x)\|^2 \right]
$$

Note that for $0 \le t \le h \le \frac{1}{8L} \le \frac{1}{8}$, it holds that $1 - e^{-2t} \ge t$. Therefore,

$$
\frac{t^2 e^{-2t}}{1 - e^{-2t}} \le \frac{t^2}{t} = t.
$$

The claim has been proved.

### D.12 A General Lemma for Time Derivative of KL

We restate the following lemma from prior work [78, Section 8.1.2]; see also [79, 6], and include a proof for completeness.

**Lemma 20** (Time derivative of KL). *Let $p_t, q_t \colon \mathbb{R}^d \to \mathbb{R}$ be probability density functions and $t \ge 0$. Suppose the following conditions hold.*

1. *There is an integrable function $\theta \colon \mathbb{R}^d \to \mathbb{R}$, i.e. $\int \theta(x) dx < \infty$, such that $\left| \frac{\partial}{\partial t} \left( p_t \ln \frac{p_t}{q_t} \right) \right| \le \theta(x)$ for all $t$ and almost every $x$.*

2. *There is an integrable function $\kappa \colon \mathbb{R}^d \to \mathbb{R}$, i.e. $\int \kappa(x) dx < \infty$, such that $\left| \frac{\partial}{\partial t} p_t \right| \le \kappa(x)$ for all $t$ and almost every $x$.*

*Then,*

$$\frac{d}{dt}\mathrm{KL}(p_t\|q_t) = \int \left(\frac{\partial p_t}{\partial t}\ln\frac{p_t}{q_t} - \frac{\partial q_t}{\partial t}\frac{p_t}{q_t}\right)dx.$$

*Proof.* We have

$$\begin{aligned}
\frac{d}{dt}\mathrm{KL}(p_t\|q_t) &= \frac{d}{dt}\int p_t\ln\frac{p_t}{q_t}\,dx \\
&= \int \frac{\partial}{\partial t}\left(p_t\ln\frac{p_t}{q_t}\right)dx \\
&= \int \left(\ln\frac{p_t}{q_t}\frac{\partial p_t}{\partial t} + p_t\frac{\partial}{\partial t}\ln\frac{p_t}{q_t}\right)dx \\
&= \int \left(\ln\frac{p_t}{q_t}\frac{\partial p_t}{\partial t} - \frac{p_t}{q_t}\frac{\partial q_t}{\partial t} + \frac{\partial p_t}{\partial t}\right)dx \\
&= \int \left(\frac{\partial p_t}{\partial t}\ln\frac{p_t}{q_t} - \frac{\partial q_t}{\partial t}\frac{p_t}{q_t}\right)dx.
\end{aligned}$$

Here, the second equality uses the Leibniz integral rule, which holds because of the first domination condition. The last step is by

$$\int \frac{\partial p_t}{\partial t}dx = \frac{d}{dt}\int p_t dx = 0$$

which applies the Leibniz integral rule on $p_t$ thanks to the second domination condition. $\qquad\square$

# E   Method and Experimental Details

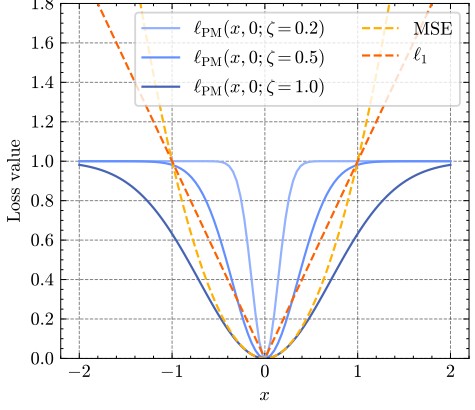

Figure 6: One-dimensional visualization of the proximal matching loss $\ell_{\mathrm{PM}}$, in comparison with the mean squared error (MSE) and mean absolute error ($\ell_1$) losses.

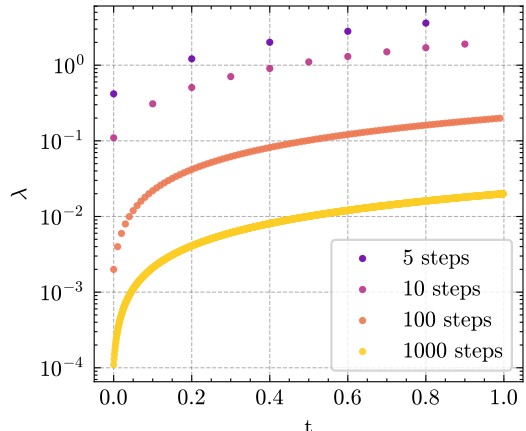

Figure 7: Values of $t$ and $\lambda$ used by `ProxDM-hybrid` under varying numbers of sampling steps. Here, we set $\beta(t) = \beta_{\min} + (\beta_{\max} - \beta_{\min})\, t$, with $\beta_{\min} = 0.1, \beta_{\max} = 20$, as used in experiments.

### E.1 Pseudo-code for Proximal Matching Training

---
**Algorithm 2** Proximal Matching Training

---
**Input:** Forward SDE noise schedule $\beta(t)$ (c.f. (1)); sampling distribution $q(t, \lambda)$ over $(t, \lambda)$; data distribution $p_0$; annealing schedule $\mathrm{Anneal}_\zeta(k)$ for $\zeta$; optimization algorithm `OptimizationStep` (e.g., SGD, Adam)

    **Initialize** network parameters $\theta_0$

    **repeat**

        $k \leftarrow k + 1$

        $\zeta_k \leftarrow \mathrm{Anneal}_\zeta(k)$                 ▷ Update $\zeta$ according to the annealing schedule

        Sample $t, \lambda \sim q(t, \lambda)$, $X_0 \sim p_0$, and $\eta, \varepsilon \sim \mathcal{N}(0, I)$ independently

        $\alpha_t \leftarrow \exp\left(-\int_0^t \beta(s)\, ds\right)$

        $X_t \leftarrow \sqrt{\alpha_t}\, X_0 + \sqrt{1 - \alpha_t}\, \eta$                     ▷ Generate $X_t \sim p_t$

        $Y \leftarrow X_t + \sqrt{\lambda}\, \varepsilon$

        Compute loss $\ell_{\mathrm{PM}} \leftarrow \ell_{\mathrm{PM}}\left(\epsilon_{\theta_k}(Y; t, \lambda),\, \varepsilon;\, \zeta_k\right)$     ▷ Proximal matching loss (cf. Eq. (9))

        $\theta_{k+1} \leftarrow$ `OptimizationStep`$(\theta_k,\, \nabla_\theta \ell_{\mathrm{PM}})$

    **until** convergence

**Output:** $f_\theta(x; t, \lambda) = x - \sqrt{\lambda}\, \epsilon_\theta(x; t, \lambda)$               ▷ Approximated $\mathrm{prox}_{-\lambda \ln p_t}(x)$

---

### E.2 Sampling $t$ and $\lambda$

We provide more details of the sampling strategy for $t$ and $\lambda$ used for proximal matching. To motivate our design, Fig. 7 displays the $(t, \lambda)$ pairs encountered by `ProxDM-hybrid` under different numbers of sampling steps: 5, 10, 100, and 1000. The $t$ values are evenly distributed in $[0, 1]$ regardless of the step count, whereas the range of $\lambda$ varies by the number of steps. Based on this observation, we design the following sampling scheme. We first sample a step number $N$ from a predefined set of candidates. Then, for the chosen $N$, we uniformly sample one of the possible $(t, \lambda)$ pairs associated with that step number. This procedure is repeated independently for each sample in the training batch and is performed on-the-fly. For `ProxDM-hybrid`, the step number candidates are $\{5, 10, 20, 50, 100, 1000\}$. For `ProxDM-backward`, we exclude 5 and 10 due to the instability associated with small step numbers. To promote balanced learning across different step numbers, we apply weighted sampling, with weights proportional to $\log(N)$ for MNIST and $N^{1/3}$ for CIFAR-10.

### E.3 Image Generation

For CIFAR-10, we adopt the same U-Net architecture as in [27]. The `ProxDM` is trained for a total of 375k iterations with a batch size of 512. We use the $\ell_1$ loss during the first 75k iterations as a pretraining step, followed by proximal matching loss with $\zeta = 2$ for 150k iterations, and then $\zeta = 1$ for the last 150k iterations. The learning rate is set to $10^{-4}$. For score models, we follow the standard practice [27, 72, 75] of using a batch size of 128, and train for 1.5M iterations to ensure the number of training samples (i.e., effective training epochs) matches that of the proximal model. We set the learning rate to $2 \times 10^{-4}$ and use gradient clipping and learning rate warm-up, following [27, 75]. For both score and proximal models, the last checkpoint is used, without model selection.

For MNIST, we halve the number of filters in the U-Net, following [72]. The `ProxDM` is trained for a total of 225k iterations with a batch size of 512, using $\ell_1$ loss with learning rate of $10^{-4}$ during the first 75k iterations, followed by proximal matching loss with $\zeta = 1$ and $\zeta = 0.5$ for 75k iterations each, with a learning rate of $10^{-5}$. For the score model, we use a batch size of 128 following the standard practice [27, 72, 75] and train for 900k iterations to ensure the number of training samples (i.e., effective training epochs) matches that of the proximal model, using a learning rate of $2 \times 10^{-4}$.

**Precision and recall metrics.** Following [38], precision measures the proportion of generated samples that fall within the manifold of real data. Specifically, real images are first embedded into a feature space via a pre-trained VGG-16 network [69]. Then, a hypersphere is defined around each feature vector with radius equal to the distance to its $k$th nearest neighbor, forming a volume that approximates the data manifold. Precision is then computed as the fraction of generated samples located within this volume, intuitively capturing sample quality and fidelity. Analogously, recall is defined as the fraction of real samples that lie within the manifold of generated samples, reflecting the diversity and data coverage of the generation.

## F Extension of `ProxDM` to Other SDEs and ODEs

In this section, we demonstrate conceptually how the proximal diffusion model can be extended to other SDEs and ODEs, such as the variance exploding SDE and probability flow ODE [75]. We leave the theoretical and empirical study of these extensions to future work.

Besides the variance preserving (VP) SDE in (1), an alternative forward process used in diffusion models is the variance exploding (VE) SDE [75], given by

$$dX_t = dW_t. \tag{38}$$

Here we omit the time-dependent diffusion term for simplicity. This corresponds to the standard Brownian motion initialized at the data distribution. The associated reverse-time SDE takes the form

$$dX_t = -\nabla \ln p_t(X_t)dt + d\bar{W}_t$$

where time flows backwards from $T$ to $0$.

Given a time grid $\{0 = t_0 < t_1 < \cdots < t_N = T\}$, applying forward discretization yields the following score-based algorithms widely used in generative models [72, 73]

$$X_{k-1} = X_k + \gamma_k \ln p_{t_k}(X_k) + \sqrt{\gamma_k} z$$

where $\gamma_k = t_k - t_{k-1}$ and $z \sim \mathcal{N}(0, I)$. Analogous to the derivation of `ProxDM` for the VP SDE, we can apply backward discretization to the reverse-time VE SDE:

$$X_{k-1} = X_k + \gamma_k \ln p_{t_{k-1}}(X_{k-1}) + \sqrt{\gamma_k} z,$$

and obtain a new proximal sampling scheme:

$$X_{k-1} = \text{prox}_{\gamma_k \ln p_{t_{k-1}}} \left( X_k + \sqrt{\gamma_k} z \right).$$

Furthermore, consider a general forward SDE of the form

$$dX_t = f(X_t, t)dt + g(t)dW_t,$$

where $f\colon \mathbb{R}^d \times \mathbb{R} \to \mathbb{R}^d$ denotes the drift coefficient and $g\colon \mathbb{R} \to \mathbb{R}$ the diffusion coefficient. [75] introduce the probability flow (PF) ODE, given by

$$dX_t = \left[ f(X_t, t) - \frac{1}{2}g(t)^2 \nabla \ln p_t(X_t) \right] dt,$$

where time flows backwards from $T$ to $0$. The PF ODE shares the same marginal probability densities as the forward process, and therefore, admits sampling algorithms analogous to those based on the reverse-time SDE. Notably, the DDIM sampler [70] has been shown to correspond to a discretization of the PF ODE [67, 53].

In particular, for the VP SDE (1), we have $f(x, t) = -\frac{1}{2}\beta(t)x$ and $g(t) = \sqrt{\beta(t)}$, which leads to the following PF ODE

$$dX_t = \left[ -\frac{1}{2}\beta(t)X_t - \frac{1}{2}\beta(t)\nabla \ln p_t(X_t) \right] dt.$$

Applying backward discretization yields

$$X_{k-1} = X_k + \frac{\gamma_k}{2}X_{k-1} + \frac{\gamma_k}{2}\nabla \ln p_{t_{k-1}}(X_{k-1}),$$

where $\gamma_k = \int_{t_{k-1}}^{t_k} \beta(s)ds$. This, in turn, gives rise to a proximal-based ODE sampler:

$$X_{k-1} = \mathrm{prox}_{-\frac{\gamma_k}{2-\gamma_k} \ln p_{t_{k-1}}} \left( \frac{2}{2-\gamma_k} X_k \right).$$

# G   Connection between Backward Discretization and Proximal Algorithms

In this section, we provide the detailed derivations for previously explored connections between proximal algorithms and backward discretization, using two continuous-time processes as examples: gradient flow and Langevin dynamics.

First, consider the gradient flow ODE

$$dx_t = -\nabla f(x_t)dt$$

whose forward (Euler) discretization gives the gradient descent algorithm

$$x^+ = x - \Delta t\, \nabla f(x)$$

where $x$ and $x^+$ denote the current and the next iterates, respectively. On the other hand, backward discretization yields the proximal point method [62]:

$$x^+ = x - \Delta t\, \nabla f(x^+) \quad \longrightarrow \quad x^+ = \mathrm{prox}_{\Delta t\, f}(x).$$

In the context of sampling, forward discretization of the Langevin dynamics

$$dX_t = -\nabla f(X_t)dt + \sqrt{2}dW_t.$$

leads to the unadjusted Langevin algorithm

$$X^+ = X - \Delta t\nabla f(X) + \sqrt{2\Delta t}z, \quad z \sim \mathcal{N}(0, I)$$

while backward discretization leads to the proximal Langevin algorithm [60, 5, 20, 79]:

$$X^+ = X - \Delta t\nabla f(X^+) + \sqrt{2\Delta t}z \quad \longrightarrow \quad X^+ = \mathrm{prox}_{\Delta t\, f}\left( X + \sqrt{2\Delta t}z \right).$$

# H   Additional Results

## H.1   Ablation on Smoothness of Target Distribution in 1D

We further investigate how the smoothness of the target distribution affects sampling performance. Specifically, we consider a one-dimensional target distribution given by $p(x) \propto (0.5 + x)^\alpha(0.5 -$

$x)^{\alpha}, x \in [-0.5, 0.5]$, with varying smoothness parameters $\alpha \in \{0, 0.5, 1.0, 2.0, 5.0\}$. This family starts from a uniform distribution over $[0, 1]$ with non-smooth boundaries ($\alpha = 0$) and becomes increasingly smooth as $\alpha$ grows. The score-based sampler uses the canonical Euler–Maruyama discretization (without the additional final denoising step), while the proximal sampler follows the hybrid update in Algorithm 1. As shown in Fig. 8, both samplers recover the target distribution when given sufficient sampling steps. The dependence of sample quality on the smoothness parameter $\alpha$ is relatively subtle—likely due to the smoothing effect inherent in the forward diffusion process. Nevertheless, ProxDM provides samples that consistently fall within the true support and achieves better alignment with the target density, especially at smaller step counts.

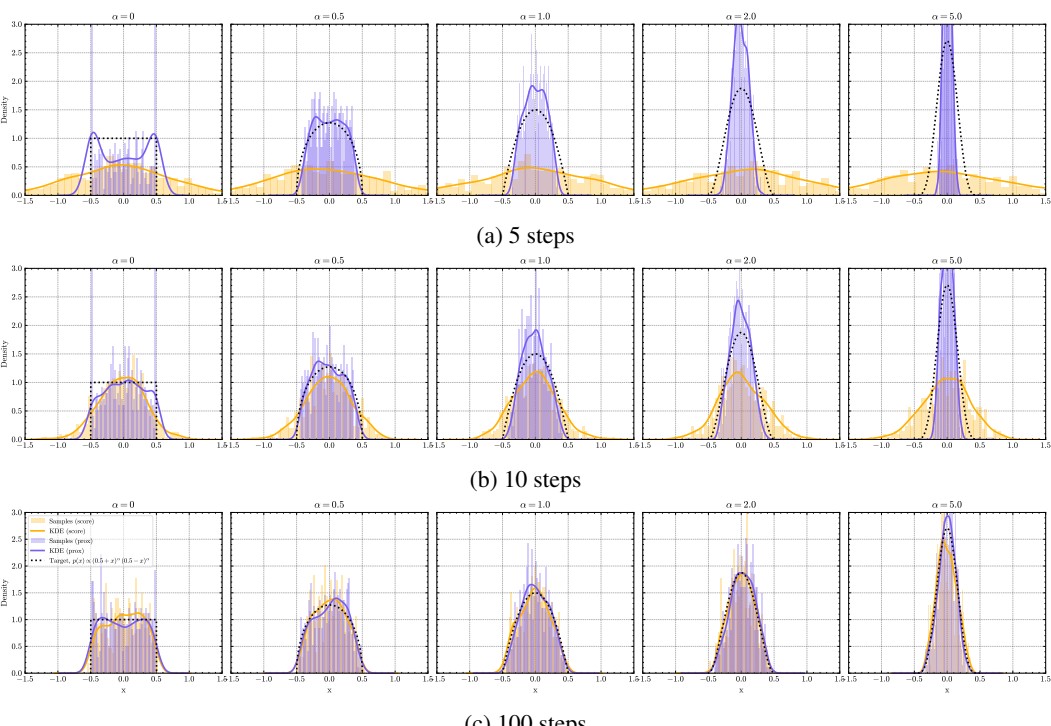

(a) 5 steps

(b) 10 steps

(c) 100 steps

Figure 8: Samples from score-based and proximal samplers using 5, 10, and 100 sampling steps for distributions with varying level of smoothness at the boundary (controlled by $\alpha$).

## H.2 Additional Results

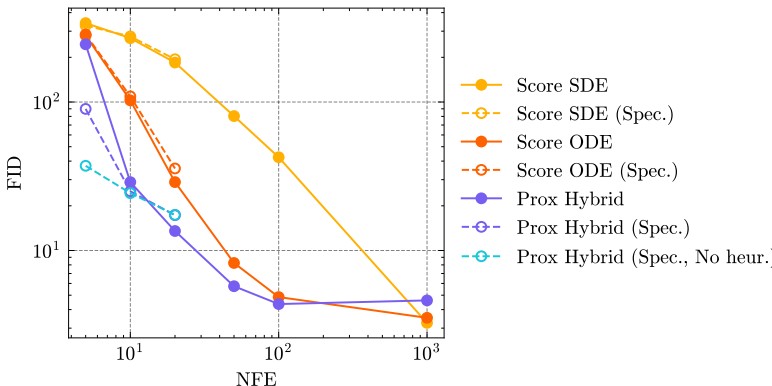

Figure 9: FID vs. number of sampling steps (or number of function evaluations, NFE) on CIFAR10. Dashed lines indicate models trained specifically for 5, 10 and 20 steps ("Spec."), while solid lines represent full-range models. The result labeled "Prox Hybrid (Spec., No Heur.)" is obtained without applying the heuristics for network parameterization and objective balancing described in Section 3.3.

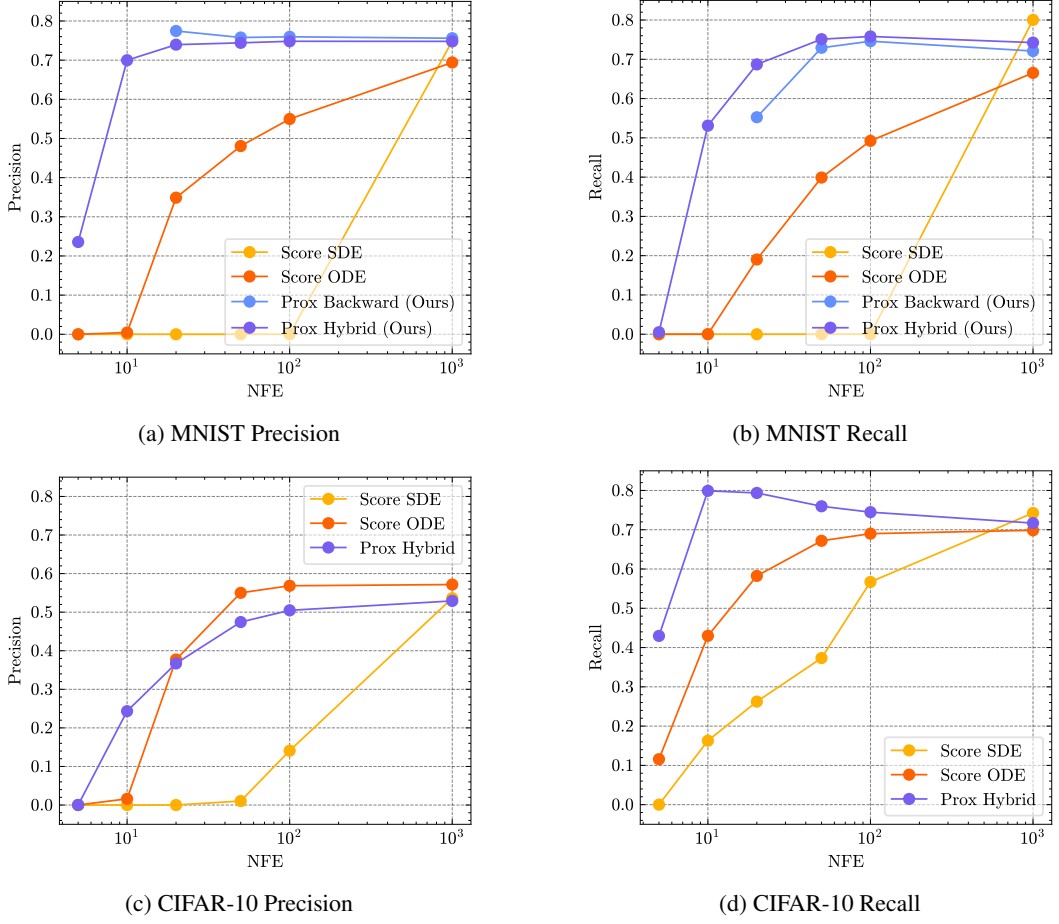

(a) MNIST Precision

(b) MNIST Recall

(c) CIFAR-10 Precision

(d) CIFAR-10 Recall

Figure 10: Precision and recall metrics [38] for MNIST and CIFAR-10 datasets. Precision measures the sample quality while recall reflects the coverage of the data distribution. Across most sampling steps (NFE), `ProxDM` achieves higher precision and recall than score-based baselines, indicating improved sample quality and diversity.

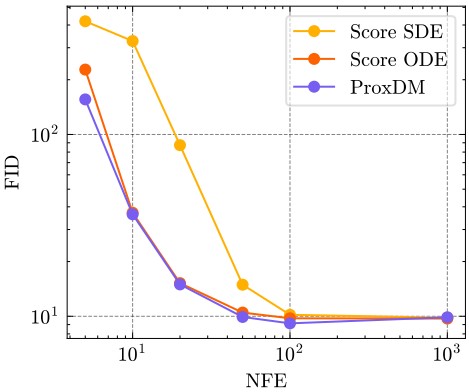

Figure 11: FID vs. number of sampling steps (NFE) on CelebA-HQ ($256 \times 256$) from score SDE, score ODE and hybrid `ProxDM` samplers. FID is computed over 30,000 generated samples.

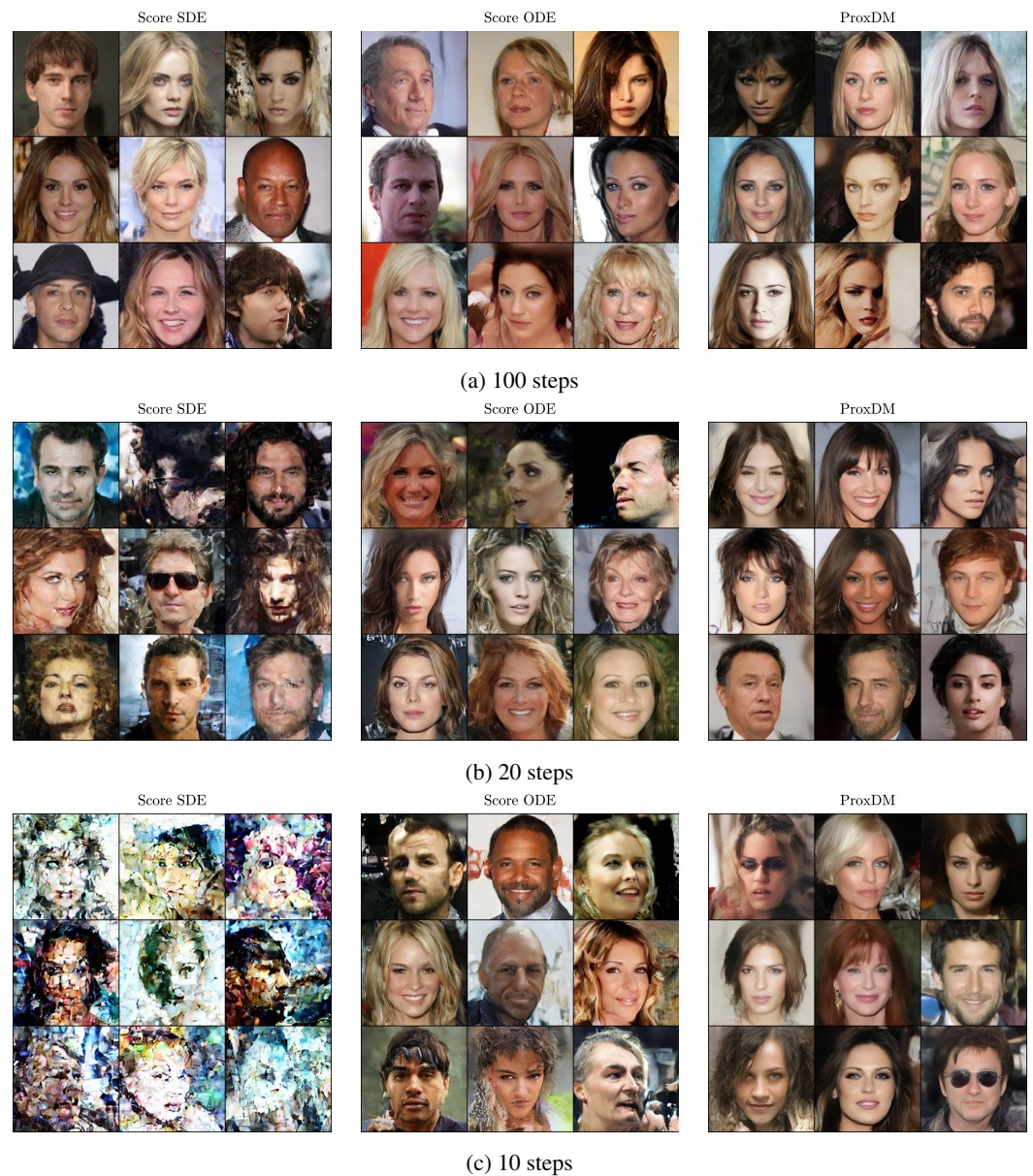

Figure 12: Uncurated CelebA-HQ ($256 \times 256$) samples from score SDE, score ODE and hybrid `ProxDM` samplers. `ProxDM` produces cleaner and more coherent samples with fewer sampling steps.

