# OpenReview forum: "Beyond Scores: Proximal Diffusion Models"
_NeurIPS.cc/2025/Conference — NeurIPS 2025 poster_

### Official Review · Reviewer_ZawX · 2025-06-28

**Clarity:** 4
**Significance:** 3
**Originality:** 3
**Rating:** 5
**Confidence:** 4

**Summary:**

This paper proposes to use proximal optimization techniques to speed up sampling in diffusion models. Two variants of the algorithm is proposed, one uses full backward discretization, and one uses a hybrid of current and previous point. Motivated by the algorithm, the authors also proposed proximal loss functions to train the "proximal score", giving the data-dependant proximal operator used during sampling. The authors analyzed the convergence properties, discussed relations to the usual forward discretization, and verified performance on synthetic and real image datasets.

**Questions:**

1. Why can't you let $\zeta$ go to zero? Is the schedule used in the CIFAR-10 experiments (which I appreciate the honesty) from previous work? How important is this parameter?

2. Why does the synthetic dataset composed of very well separated points? What if you use a distribuion where the outline of the picture is more connected, i.e. almost on a low-d manifold? This would provide a case where the score field becomes low-d, and hence help show advantage to MMSE learning?

3. Why can't the FID at high function evaluation beat the normal forward diffusion? Please discuss on this, as people still care about the quality.

4. Why is the backward method missing in the CIFAR-10 experiment?

**Ethical Concerns:**

["NO or VERY MINOR ethics concerns only"]

**Final Justification:**

It's a good paper. Sound techniques and theory, and full of useful details. The clarifications from this discussions should improve the paper. The only downside that prevents a 6 from me is empirical performance being not super impressive. But other than that, this paper should get in.

**Limitations:**

Limitations are well-discussed. Although addressing the performance saturation at high NFE would be helpful.

**Paper Formatting Concerns:**

None.

**Quality:**

4

**Strengths And Weaknesses:**

Strength:
1. The method is well-motivated by research in proximal-optimization, and is well-developed into two plausible versions.
1. The sampling method then motivates a new loss function, leading to the a new proximal score matching technique that is novel to my knowledge.
2. Writing of the paper is of exceptionally high quality. The main text is accessible to a wide audience, has the perfect amount of technicality, with connections to more conventional methods clearly discussed. I did not fully checked the proofs in Appendix (yet).
3. The experimental results show promising performance at small number of function evaluations. I cannot wait to see more adoption of this method.
4. I also appreciate that the authors spent enough passage discussing implementation details, including the crucial reparameterization of the model. I find these non-trivial details very useful and educational, highlighting the practical focus of the authors, in addition to their analytical rigor.

Weaknesses:
There is hardly any weaknesses of large concerns (except the proofs in Appendix which I haven't checked). I point out a few minor suggestions.
1. The authors discussed the benefits of the MAP estimate from proximal points vs the MMSE denoiser (which is provides a lot of nice insights). However, the experiments did not support the claims. I would have thought that the synthetic dataset has a sharp score field at low noise, and hence the authors would show that the good performance comes from using the MAP instead of MMSE. It would be great to use this experiment to further develop this point.
2. I am a little confused by the definition of $f_\theta(\cdot;t,\lambda)$, I think the reader can figure it out almost by themselves, but it would be nice to provide a quick derivation (e.g. in the extra page if accepted).
3. Would be nice to provide some intuition on the loss of Fang et al., if there is any. I think this is a very cool reference to read, but for self-contained purposes do give wide audience some high-level ideas.
4. It would be nice to visualize the score field of the proximal model. I think this would help us understand the differences and connections between the two frameworks, and give readers more visuals to help understanding and interpreting the method.

If I have time I will read more into the Appendix, and provide more feedback. I put my confidence low for now (sorry for the limited review). If the authors could address the points above and provide more results (the suggestions do not require any large-scale experiments on new datasets), I am willing to further increase the score.

---

> ### Author Rebuttal · Authors · 2025-07-31
>
> - **”There is hardly any weaknesses of large concerns (except the proofs in Appendix which I haven't checked). I point out a few minor suggestions.”**
>
> **Response:** We are glad the reviewer enjoyed our work!
>
>
> - **”The authors discussed the benefits of the MAP estimate from proximal points vs the MMSE denoiser (which is provides a lot of nice insights). However, the experiments did not support the claims. I would have thought that the synthetic dataset has a sharp score field at low noise, and hence the authors would show that the good performance comes from using the MAP instead of MMSE. It would be great to use this experiment to further develop this point.”**
>
> **Response:** We appreciate the reviewer’s interest in the MAP vs. MMSE comparison. Could the reviewer please clarify what aspect of the experiment they found unconvincing? To clarify, as shown in Figure 1, our ProxDM method - based on MAP denoisers - outperforms the score samplers based on MMSE denoiser, in terms of coverage of the target distribution and Wasserstein distance. Please note that no training is involved in the synthetic experiment - we use the exact score and proximal operators that were analytically derived from the ground-truth distribution.
>
> Prior work has shown that MAP denoisers produce more perceptually appealing outputs than MMSE denoisers [1], which conceptually supports our use of proximal operators in generative models.
>
> We are happy to further develop these experiments as the reviewer suggests.
>
> [1] Blau, Yochai, and Tomer Michaeli. "The perception-distortion tradeoff." In Proceedings of the IEEE conference on computer vision and pattern recognition, pp. 6228-6237. 2018.
>
>
>
> - **”I am a little confused by the definition of f_theta(;t, lambda), I think the reader can figure it out almost by themselves, but it would be nice to provide a quick derivation (e.g. in the extra page if accepted).”**
>
> **Response:** Certainly, we are happy to add more clarification on this. We use $f_\theta( ; t, \lambda)$ to denote the learned proximal operator at time $t$ and with regularization parameter $\lambda$, i.e., $prox_{-\lambda \ln p_t}$. In implementation, we parameterize $f$ by $f_\theta(x; t, \lambda) = x - \epsilon_\theta(x; t, \lambda)$, where $\epsilon_\theta$ is a neural network. We found this leads to better convergence of our model at large step numbers in early experiments. This parameterization is inspired by a similar $\epsilon$ parameterization in DDPM (please see equation 11 in [1]).
>
> [1] Ho, Jonathan, Ajay Jain, and Pieter Abbeel. "Denoising diffusion probabilistic models." NeurIPS 2020.
>
>
> - **”Would be nice to provide some intuition on the loss of Fang et al., if there is any. I think this is a very cool reference to read, but for self-contained purposes do give wide audience some high-level ideas.”**
>
> **Response:** Certainly, we can add more intuition in the text, with a figure illustrating the loss function. Useful intuition comes from trying to recover the mode of a distribution in 1D. In that case, an L0 pseudo-norm (if one could minimize such a non-differentiable loss function), recovers the maximum of a distribution. The proximal matching loss is a generalization of this idea to R^d and differentiable functions.
>
>
> - **“It would be nice to visualize the score field of the proximal model. I think this would help us understand the differences and connections between the two frameworks, and give readers more visuals to help understanding and interpreting the method.”**
>
>
> **Response:** Thank the reviewer for this suggestion. Can we ask what the reviewer meant by the “score field of the proximal model”? If it refers to the displacement field of prox (i.e., prox(x) - x), we are happy to add such a comparison in 2D space. However, please note that we cannot include images in the rebuttal.
>
>
>
> - **”Why can't you let \zeta go to zero? Is the schedule used in the CIFAR-10 experiments (which I appreciate the honesty) from previous work? How important is this parameter?”**
>
> **Response:** We thank the reviewer for this question. $\zeta$ is the parameter in the proximal matching loss, and taking it to zero is the correct approach from a theoretical standpoint. In practice, however, when $\zeta$ equals zero, the loss can blow up to infinity. In experiments, we use a schedule to set $\zeta$, which can be tuned for different datasets and tasks. Previous work does not provide a schedule for image generation on CIFAR-10.  In our early experiments, we tuned hyperparameters for $\zeta$, e.g., extreme values and decay rate. Although theoretically speaking, a smaller $\zeta$ brings the learned mapping closer to the true proximal operator, empirically, a smaller $\zeta$ does not always lead to improved generation performance. In general, we found that $\zeta$ between 0.5 and 1 leads to similar performance in the tasks we considered. However, we also observed that the optimal $\zeta$ could vary based on the dataset and even the number of sampling steps—a more detailed study of the relation between $\zeta$ and generation performance is an interesting direction for future work.
>
>
> - **”Why does the synthetic dataset composed of very well separated points? What if you use a distribution where the outline of the picture is more connected, i.e. almost on a low-d manifold? This would provide a case where the score field becomes low-d, and hence help show advantage to MMSE learning?”**
>
> **Response:** Thank the reviewer for this suggestion. The synthetic dataset we use comes directly from the DataSauraus dataset [1]—we did not perform any explicit subsampling. We are working on adding an experiment on a low-dimensional manifold comparing MAP and MMSE learning.
>
> [1] Matejka, Justin, and George Fitzmaurice. "Same stats, different graphs: generating datasets with varied appearance and identical statistics through simulated annealing." In Proceedings of the 2017 CHI conference on human factors in computing systems, pp. 1290-1294. 2017.
>
>
>
> - **”Why can't the FID at high function evaluation beat the normal forward diffusion? Please discuss on this, as people still care about the quality.”**
>
> **Response:** Thank the reviewer for pointing this out. We argue that this is not a significant limitation: 1) Practically speaking, using 1000 time steps is usually computationally expensive and impractical for many applications. Our method is designed to be more efficient while still achieving competitive results. 2) The difference between the FID in this regime does not lead to significant differences in the visual quality of the samples. In contrast, our method can generate more visually appealing samples with fewer steps. 3) Improving performance in the fine-grained discretization regime is not the primary focus of this work. Our emphasis is on high quality under low step budgets. 4) Additionally, although our algorithm requires a proximal operator at each step, the mapping learned by our U-Net is not explicitly constrained to be a proximal (but only asymptotically). In the future, we believe that improved network parameterization (such as that from [1]) and training strategies may close the gap at high step counts.
>
> [1] Fang et al. "What's in a prior? learned proximal networks for inverse problems." ICLR 2024.
>
>
>
>
> - **”Why is the backward method missing in the CIFAR-10 experiment?”**
>
>
> **Response:** Thank the reviewer for raising this point. The backward method has a practical limitation on the maximum step size it can take. As shown in Eq. (PDA) and Algorithm 1, each backward update involves the proximal operator $prox_{-\lambda \ln p_{ t_{k-1} } }$, where the regularization weight is $\lambda = \frac{2\gamma_k}{2 - \gamma_k}$. To ensure $\lambda > 0$, we require $\gamma_k < 2$, and because $\gamma_k$ increases with step size, this imposes a hard upper bound on the step size. In practice, as the step size increases, $\gamma_k$ approaches 2, and the update becomes increasingly unstable, which degrades sampling quality at few steps. On MNIST, we found performance degrades noticeably at around 20 steps. On CIFAR-10, we observed the instability issue become more pronounced, which led to poor performance at few steps. For this reason, we chose not to include the backward method in the CIFAR-10 experiment. That said, we plan to explore ways to address this limitation in future work.

---

> > ### Comment · Reviewer_ZawX · 2025-08-04
> > **Further questions**
> >
> > I thank the authors for their detailed reply. They have addressed some of the concerns. I hope the authors could address those points in a revision, and include the results / figures they promised.
> >
> > I still wanted to push a bit on the following points:
> >
> > **Advantage of MAP and MMSE**
> > I was referring to the advantage of in lines 111-113, that the MAP denoising is agnostic to the smoothness / differentiability of the score. The experiment with sharp features was referring to the one done on DataSauraus. I don't think any of the experiments could confirm that on this dataset where the points are on a low-d manifold that the proposed method (more related to the MAP denoising) realises the promised advantage. The authors already said that they are working on an experiment to show this: will this result be included in the final version? If so, what is the hypothesis? And can the authors share some details? Despite that no figures is allowed, some of this details are important for the reviewers.
> >
> > **Worse FID on 1000 steps**
> > I did not raise this as a limitation (I don't particularly care about driving FIDs down), but I do think that results like this deserve some explanations. There are maybe a few quasi-equivalent ways of asking the same question, to make it clear to the authors:
> > a. If there are some theory suggsting that the advantage of the the proposed method dimishes with more sampling steps?
> > b. If there is no inherent limitation of this method compared to previous methods, how come this happens?
> > From my reading of the paper, I don't believe there is a strong reason to expect such degradation at this regime. This makes such result surprising to readers.
> >
> > Further, I also believe that 1000 steps isn't a rediculous amount of sampling steps. For the field I'm working on, running 1000 steps is far from computationally demanding. I hope the authors could address this point directly instead of casting it away; and this point is also raised by reviewer uiu807.
> >
> > **Residual parameterization**
> > I am still not sure if the parameterization $f_\theta(\cdot; t, \lambda) = \cdot - \epsilon_\theta(\cdot, ; t, \lambda) $ is well explained. "The residual of the proximal operator" is also not clearly defined. I am guess now that this is just from an rearrangement from Eqn (6). Is this correct? Or is "The residual of the proximal operator" defined elsewhere?
> >
> > (I apologize for the typos, which I admit was the result of a rushed review).

---

> > > ### Author Response · Authors · 2025-08-05
> > > **Response to Further Questions**
> > >
> > > - **Advantage of MAP and MMSE**
> > >
> > > Thank the reviewer for the clarification. We plan to add an experiment to study the sampling performance as a function of the smoothness of the target distribution. Specifically, we can sample from a finitely-supported target distribution in 1D with a density that is increasingly non-smooth at the boundary. One example is $p(x) \propto \mathbb{1}_{[-0.5, 0.5]} \cdot (0.5 + x)^\alpha \cdot (0.5 - x)^\alpha$, where changing $\alpha$ changes the smoothness of the function at the boundary. We will study the performance of ProxDM and DDPM under different $\alpha$'s and report these results in the paper. If no significant difference is observed, we will appropriately adjust our statement. Please let us know if this proposed experiment addresses your question.
> > >
> > >
> > > - **Worse FID on 1000 steps**
> > >
> > > We thank the reviewer for the clarification. Our conjecture as to what might be driving the lack of FID going down at high sampling steps is that ProxDM, unlike score-based methods, relies on solving an implicit equation. In practice, we use proximal matching to train networks that are "close" to proximal operators, but they are likely not **exact** proximal operators. As a result, the computed updates violate the conditions that define such updates, introducing inaccuracies. We believe that these inaccuracies tend to be more important (at high sampling steps) than the inaccuracies in the learned score from DDPM. We will add this as a discussion in the paper.
> > >
> > >
> > > - **Residual parameterization**
> > >
> > > We thank the reviewer for pointing out the confusion. By "The residual of...", we mean the difference between the output and input of a function. In particular, we define the (learned) proximal operator as $f_\theta( x ; t, \lambda)$ and its residual as $\epsilon_\theta( x ; t, \lambda) = [ x - f_\theta( x ; t, \lambda) ] / \sqrt{\lambda} $. During training, instead of directly parameterizing $f_\theta( \cdot ; t, \lambda)$ by a neural network, we parameterize its residual $\epsilon_\theta( x ; t, \lambda)$ by a neural network, and then compute the proximal operator by $f_\theta( \cdot ; t, \lambda) = \cdot - \sqrt{\lambda} \epsilon_\theta( \cdot ; t, \lambda)$. This parameterization led to better results at large step numbers in our early experiment. We will modify Section 3.3 to make this clearer.

---

> > > > ### Comment · Reviewer_ZawX · 2025-08-06
> > > > **Thank you**
> > > >
> > > > The answers address all questions. I congratulate the authors on the high quality of the idea, results and also written quality. If the clarifications and details from this discussion get into the final version, it will make the paper ready to impress.

---

### Official Review · Reviewer_JaE7 · 2025-07-01

**Clarity:** 4
**Significance:** 4
**Originality:** 3
**Rating:** 5
**Confidence:** 3

**Summary:**

The paper proposes an alternative approach to training diffusion models - based on *proximal matching* instead of score matching. The key idea behind their approach is to use a *backward discretization* of the reverse stochastic process  for the sampling (in place of the standard forward discretization used by score matching methods).

They show that the backward discretization update can be interpreted as a proximal step w.r.t the log-density, and provide an algorithm "ProxDM" (with two variants for two kinds of discretizations) to generate samples using this proximal operator (analogous to the sampling algorithm used by score-based models like DDPMs).

They show how to train a neural network to learn this proximal step ("proximal-matching"). Their method involves sampling different times and regularizer strengths (t, $\lambda$ pairs) in order to to handle multiple step sizes at sampling time, and they also propose appropriate parametrization and loss weighting to balance the training loss contribution of different times / regularizer strengths.

They also prove (under assumption that training algorithm converges to oracle proximal operator) that both variants of their ProxDM sampling algorithm converge to the true distribution, at a faster rate (fewer sampling steps) than score-matching methods.

Finally, they provide experimental verification using toy datasets (MNIST and CIFAR10) that ProxDM generates better samples than score-matching methods for small number of steps.

**Questions:**

(nit) The text says that you include pseudo-code for proximal-matching training in the appendix but I couldn't find it, could you please add it (or else remove this statement)?

My main questions are about comparison with score-based methods:

1. [Sampling of (t, $\lambda$) pairs during training]: since the choice of $\lambda$ depends on the number of sampling steps N that we wish to use during inference, ProxDM requires training the network with, say, 10x more ( t, $\lambda$  ) pairs in order to support 10 different Ns compared to a single fixed N. Is this an inherent limitation of ProxDM compared to score-based methods like DDPM (which as I understand requires sampling only t during training, and then any number of sampling steps can be used at inference)?

2. For score-matching, the training objective is actually equivalent to minimizing the variational bound on the negative log likelihood of the training data. Does the ProxDM training objective also have such an equivalent formulation?

**Ethical Concerns:**

["NO or VERY MINOR ethics concerns only"]

**Final Justification:**

The authors have done an excellent job with this paper. They also acknowledges my concerns satisfactorily and therefore I have kept my "Accept" rating.

**Limitations:**

Yes

**Quality:**

3

**Strengths And Weaknesses:**

Strengths:
1. Excellent presentation, nice theory, very easy and enjoyable read especially for a mathematically dense paper.
2. The idea of using proximal operators to do sampling in diffusion models is certainly novel, and the motivation behind it (faster convergence) is very compelling.
3. The authors clearly explain the salient differences between their method and the score-based method, and also provide convenient mental models for comparing them (e.g. interpretation as MAP denoiser vs MMSE denoiser).
4. The paper does a great job closing the loop from theory to practically useful methods (evidenced by their detailed analysis/experiments which revealed that the hybrid discretization method is better in practice even though the fully backward discretization was better in theory). They didn't shy away from discussing practical considerations (sampling of t and lambda, loss weighting) and limitations (minimum steps requirement for the fully backward method).

Weaknesses:
More experimental results could have been provided (e.g. using large image datasets like CelebA) to better justify the practical utility of these methods for "very high quality" image generation. In this regime (large image + high quality), I would have liked to know if either of the two ProxDM variants give a significant reduction in number of steps compared to score-based methods. It would also be interesting to see if the theoretical minimum on number of steps for the fully backward discretization is even a concern in this regime.

---

> ### Author Rebuttal · Authors · 2025-07-31
>
> - **”More experimental results could have been provided (e.g. using large image datasets like CelebA) to better justify the practical utility of these methods for "very high quality" image generation. In this regime (large image + high quality), I would have liked to know if either of the two ProxDM variants give a significant reduction in number of steps compared to score-based methods. It would also be interesting to see if the theoretical minimum on number of steps for the fully backward discretization is even a concern in this regime.”**
>
> **Response:** Thank you for this suggestion. Yes, the empirical component of our work could have been more comprehensive (we focused mainly on the foundational components of the work, and experimentally we only got to perform experiments on CIFAR). We are implementing a latent space version of our model for generation of high-dimensional images, as suggested.
>
>
> - **”The text says that you include pseudo-code for proximal-matching training in the appendix but I couldn't find it, could you please add it (or else remove this statement)?“**
>
> **Response:** Certainly. We're sorry about that. We will certainly add it.
>
>
> - **Question 1: [Sampling of (t, lambda) pairs during training]: since the choice of depends on the number of sampling steps N that we wish to use during inference, ProxDM requires training the network with, say, 10x more (t, lambda) pairs in order to support 10 different Ns compared to a single fixed N. Is this an inherent limitation of ProxDM compared to score-based methods like DDPM (which as I understand requires sampling only t during training, and then any number of sampling steps can be used at inference)?”**
>
> **Response:** The reviewer is correct. ProxDM requires training with different sets of (t, lambda) pairs to support different sampling steps, because the proximal operator invoked at each step depends on the step size. Score models natively support arbitrary sampling steps, as the score function invoked at each step is independent of step size. We plan to explore continuous sampling strategies for (t, lambda) in the future to train any-step ProxDM.
>
>
> - **Question 2: “For score-matching, the training objective is actually equivalent to minimizing the variational bound on the negative log likelihood of the training data. Does the ProxDM training objective also have such an equivalent formulation?”**
>
> **Response:**
> Thank you for this insightful question.
>
> 1) Since our model is mainly built upon the continuous-time SDE framework in [1], it does not naturally come with an MLE interpretation as originated from the discrete framework in DDPM [2]. It remains an open question whether such an equivalence exists between proximal matching and MLE training, and establishing such an equivalence can be non-trivial. For instance, in the derivation in [2], the KL term between Gaussian distributions reduces to an MSE between means—this is critical for linking denoising score matching to ELBO. In contrast, our proximal matching loss is not an MSE, and the transition kernel in ProxDM is non-Gaussian. Nonetheless, we believe this is a promising direction and plan to explore it in future work.
>
> 2) Although our method does not have a likelihood-based guarantee, it is supported by rigorous theoretical guarantees from a sample distribution perspective. In Theorem 1, we showed that if true proximal operators are known, the sample distribution generated by our algorithm converges to the true data distribution in KL divergence. Furthermore, prior work [3] establishes theoretical guarantees for learning true proximal operators via proximal matching. Together, these results provide strong theoretical support for our approach.
>
> 3) Additionally, as Gribonval [4] showed, MAP and MMSE denoisers can be closely related under a change of distributions. This suggests that our training paradigm may correspond to MLE training for a related distribution, which may be worth investigating in the future.
>
>
> References:
>
>
> [1] Song, et al. "Score-based generative modeling through stochastic differential equations." ICLR 2021.
>
> [2] Ho, et al. "Denoising diffusion probabilistic models." NeurIPS 2020.
>
> [3] Fang, et al. "What's in a prior? learned proximal networks for inverse problems." ICLR 2024.
>
> [4] Gribonval R. “Should Penalized Least Squares Regression be Interpreted as Maximum A Posteriori Estimation?” IEEE Trans. Signal Process. 2011.

---

> > ### Comment · Reviewer_JaE7 · 2025-08-06
> >
> > Thank you for answering my questions, I'm very satisfied with the responses and would like to keep my Accept rating.

---

> ### Comment · Area_Chair_RMhK · 2025-08-05
>
> Dear reviewer,
> If not already, please take a look at the authors' rebuttal, and discuss if necessary.
> Thanks,
> -AC

---

### Official Review · Reviewer_ot8V · 2025-07-05

**Clarity:** 3
**Significance:** 3
**Originality:** 3
**Rating:** 5
**Confidence:** 4

**Summary:**

This paper proposes a new method for improving the discretization for backwards processes for diffusion models. Inspired by proximal techniques in optimization, they derive a proximal method-based solver for the backwards process of the OU evolution, as well as a hybrid method, loosely based on forward-backwards methods in optimization theory. Their main theoretical result is a set of error bounds for these methods, given L2-accurate proximal operators. Under some reasonable assumptions on the distribution, they show that the proximal method achieves error eps with O(d / eps) steps, and the hybrid method requires O(d / sqrt(eps)).

**Questions:**

One feedback I have is that it would be nice to have a more complete comparison to the previous non-asymptotic rates for how many timesteps are necessary, given L2 accurate score functions. I believe the best rates are currently given by [1]; notably, their dependence on their eps is 1/eps^2, which maybe worth noting (although again these results are not quite comparable).

[1] Nearly d-Linear Convergence Bounds for Diffusion Models via Stochastic Localization. Benton et al., ICLR 2024.

**Ethical Concerns:**

["NO or VERY MINOR ethics concerns only"]

**Final Justification:**

The authors resolved my concerns to my satisfaction. I still believe the paper is quite strong and I will maintain my original score.

**Limitations:**

yes

**Quality:**

3

**Strengths And Weaknesses:**

Quality: The technical content and ideas presented in the paper are very nice. The approach taken by the paper is very natural in retrospect given the success of proximal methods for optimization, and the bounds they obtain are relatively clean, and assuming learning accurate proximal estimators is about as difficult as learning score functions, achieve better error rates in terms of eps, although of course this is not directly comparable.

Clarity: The paper is well-written and very easy to read.

Significance: This method in this paper could yield new, improved algorithms for solvers for diffusion models, which are a very important area of study.

Originality: The idea of using such proximal methods, while clearly inspired by optimization, is very nice in this setting.

---

> ### Author Rebuttal · Authors · 2025-07-31
>
> - **“One feedback I have is that it would be nice to have a more complete comparison to the previous non-asymptotic rates for how many timesteps are necessary, given L2 accurate score functions. I believe the best rates are currently given by [1]; notably, their dependence on their eps is 1/eps^2, which maybe worth noting (although again these results are not quite comparable).”**
>
> [1] Nearly d-Linear Convergence Bounds for Diffusion Models via Stochastic Localization. Benton et al., ICLR 2024.
>
> **Response:** Thank you for this comment. Yes, we are happy to add this comparison and a new paragraph on more comprehensive comparison to previous works. We agree with the reviewer that their results are not directly comparable to ours: they assumed non-smooth data distributions and L2-accurate score estimates, whereas we assumed L-smooth distributions and exact proximal operators. We are working on extending our analysis to learned proximal operators and non-smooth distributions.

---

### Official Review · Reviewer_uiu8 · 2025-07-07

**Clarity:** 3
**Significance:** 2
**Originality:** 3
**Rating:** 4
**Confidence:** 4

**Summary:**

Proximal Diffusion Models propose learning the proximal operator of the reverse‐time SDE in diffusion generative models, rather than estimating its score (gradient of the log probability). This operator directly produces the MAP denoised output under a Gaussian corruption model, simplifying each sampling step. The authors introduce a “proximal matching” objective to train this network and back it up with rigorous convergence guarantees, showing that Proximal Diffusion Models reaches high‐quality samples in fewer iterations than traditional score‐based methods.

**Questions:**

- Proximal matching trains a MAP denoiser under Gaussian corruption (i.e., effectively targeting the mode of the posterior) whereas score‐based diffusion methods optimize a variational bound that (at least loosely) increases the model’s log‐likelihood, this can be more explicit with MLE diffusion weighting. Do you have any theoretical guarantees that minimizing the proximal matching loss will also improve—or even converge to—the true data log‐likelihood? In other words, is there an ELBO‐style bound or other analysis showing that training with a MAP denoiser yields maximum‐likelihood consistency?
- The paper reports faster convergence, but is that improvement limited to the sampling phase, or does it extend to faster convergence during training as well?
- With current trends of one-pass models such as consistency models, distillation methods, shortcut models, and flow models, what are the immediate benefit of proximal diffusion models? Are the main benefits faster inference? If so, those aforementioned methods already "straighten" the gradient trajectory—whether along the denoising path or the velocity field, allowing for a single one pass inference rendering a proximal operator unnecessary.
- Overall, the paper is well written and engaging, but unless you can demonstrate potential innovation for future diffusion works or significant benefits beyond sampling speed, which many recent methods already address, I remain a little skeptical, though I am open to further discussion.

**Ethical Concerns:**

["NO or VERY MINOR ethics concerns only"]

**Final Justification:**

Written in response.

**Limitations:**

Yes

**Quality:**

3

**Strengths And Weaknesses:**

Strengths
- The paper is well-structured and easy to follow.
- The derivations are comprehensive and, to the best of my knowledge, mathematically sound.
- The proofs are rigorous, and the theoretical convergence guarantees are clearly presented.
- The experiments compellingly demonstrate superior sample quality with fewer sampling steps.

Weaknesses
- Beyond sampling efficiency, no clear advantage over standard diffusion models.
- Experiments rely solely on FID versus sampling steps, offering a limited view of performance.
- On larger datasets (e.g., CIFAR-10), traditional diffusion at full timesteps still achieves better FID than the proximal approach.
- Comparative analyses of training efficiency and convergence behavior are absent.

---

> ### Author Rebuttal · Authors · 2025-07-31
>
> - **On the Significance of Our Work:**
>   - **”Beyond sampling efficiency, no clear advantage over standard diffusion models.”**
>   - **”With current trends of one-pass models such as consistency models, distillation methods, shortcut models, and flow models, what are the immediate benefit of proximal diffusion models?...”**
>
>
> **Response:** Thank you for allowing us to elaborate on this critical point. All diffusion models (both vanilla and accelerated versions) rely on *forward discretization* of continuous processes. It is true that in the last ~5 years, a lot of research has gone into developing accelerated methods based on deterministic processes (probability flows), shortcuts, distillation models, and more. Our goal was not just to provide an approach to accelerate inference using the well-established score-based paradigm, but rather to develop a new kind of diffusion model based on novel discretization schemes (which have stronger theoretical properties). A key contribution of our work is to show, for the first time, how backward discretization schemes can lead to a new type of diffusion models (based on proximal operators instead of scores). In doing so, we show that sampling based on backward discretization leads (immediately, and somewhat surprisingly) to better sampling efficiency than vanilla SDE score models. Naturally, one can envision developing corresponding accelerated models based on Prox-schemes (on ODES, shortcut methods, and more). These are only conceivable given the formalization of ProxDM.
>
> Furthermore, our work can be seen as a complementary and orthogonal line of work to these recent advancements. Our approach does not rely on any distillation technique and is trained purely from data. Moreover, the idea of backward discretization can be extended to other processes, such as the “straighter” ones mentioned by the reviewer. We believe it opens up promising directions for future research. As reviewer ZawX writes, "I cannot wait to see more adoption of this method," and reviewer ot8V, “This method in this paper could yield new, improved algorithms for solvers for diffusion models”.
>
> Finally, we believe the findings from ProxDM may lead to a novel understanding of current methods. For example, recent few-step generative models have found the benefit of MAP-promoting losses [1,2], and our findings may provide new insights into the effectiveness of these new losses and inspire new improvements in methodology.
>
> References:
>
> [1] Yang Song and Prafulla Dhariwal.  Improved techniques for training consistency models.  In International Conference on Learning Representations (ICLR), 2024.
>
> [2] Geng, Zhengyang, Mingyang Deng, Xingjian Bai, J. Zico Kolter, and Kaiming He. "Mean flows for one-step generative modeling."
>
>
>
> - **”Experiments rely solely on FID versus sampling steps, offering a limited view of performance.”**
>
> **Response:** Thank the reviewer for raising this point. Although FID is the most commonly used metric for unconditional image generation, we agree that additional metrics can provide a better view of the performance - we compute Precision and Recall and present the results in the tables below.
>
> **MNIST:**
>
> Precision:
> | Method \ NFE   |    5 |   10 |   20 |   50 |   100 |   1000 |
> |----------------|------|------|------|------|-------|--------|
> | Score SDE      | 0.00 | 0.00 | 0.00 | 0.00 |  0.00 |   0.75 |
> | Score ODE      | 0.00 | 0.00 | 0.35 | 0.48 |  0.55 |   0.69 |
> | Prox Backward  |      |      | 0.77 | 0.76 |  0.76 |   0.76 |
> | Prox Hybrid    | 0.24 | 0.70 | 0.74 | 0.74 |  0.75 |   0.75 |
>
> Recall:
> | Method \ NFE   |    5 |   10 |   20 |   50 |   100 |   1000 |
> |----------------|------|------|------|------|-------|--------|
> | Score SDE      | 0.00 | 0.00 | 0.00 | 0.00 |  0.00 |   0.80 |
> | Score ODE      | 0.00 | 0.00 | 0.19 | 0.40 |  0.49 |   0.67 |
> | Prox Backward  |      |      | 0.55 | 0.73 |  0.75 |   0.72 |
> | Prox Hybrid    | 0.01 | 0.53 | 0.69 | 0.75 |  0.76 |   0.74 |
>
> **CIFAR-10:**
>
> Precision:
> | Method \ NFE   |    5 |   10 |   20 |   50 |   100 |   1000 |
> |----------------|------|------|------|------|-------|--------|
> | Score SDE      | 0.00 | 0.00 | 0.00 | 0.01 |  0.14 |   0.54 |
> | Score ODE      | 0.00 | 0.02 | 0.38 | 0.55 |  0.57 |   0.57 |
> | Prox Hybrid    | 0.00 | 0.24 | 0.37 | 0.47 |  0.50 |   0.53 |
>
> Recall:
> | Method \ NFE   |    5 |   10 |   20 |   50 |   100 |   1000 |
> |----------------|------|------|------|------|-------|--------|
> | Score SDE      | 0.00 | 0.16 | 0.26 | 0.37 |  0.57 |   0.74 |
> | Score ODE      | 0.12 | 0.43 | 0.58 | 0.67 |  0.69 |   0.70 |
> | Prox Hybrid    | 0.43 | 0.80 | 0.79 | 0.76 |  0.74 |   0.72 |
>
> On MNIST, both variants of ProxDM consistently outperform score-based samplers in both precision and recall across most step counts from 5 to 100.
> On CIFAR-10, ProxDM shows clear advantages over Score SDE across nearly all step numbers and metrics, except for NFE=1000. Compared to the Score ODE sampler, ProxDM provides notable improvements at small NFEs (5, 10 and 20), while maintaining comparable performance at higher step numbers.
>
>
> - **”On larger datasets (e.g., CIFAR-10), traditional diffusion at full timesteps still achieves better FID than the proximal approach.”**
>
> **Response:** This is correct, and we argue that this is not a significant limitation: 1) Practically speaking, using 1000 time steps is usually computationally expensive and impractical for many applications. Our method is designed to be more efficient while still achieving competitive results. 2) The difference between the FID in this regime does not lead to significant differences in the visual quality of the samples. In contrast, our method can generate more visually appealing samples with fewer steps. 3) Improving performance in the fine-grained discretization regime is not the primary focus of this work. Our emphasis is on high quality under low step budgets. 4) Additionally, although our algorithm requires a proximal operator at each step, the mapping learned by our U-Net is not explicitly constrained to be a proximal (but only asymptotically). In the future, we believe that improved network parameterization (such as that from [1]) and training strategies may close the gap at high step counts.
>
> [1] Fang et al. "What's in a prior? learned proximal networks for inverse problems." ICLR 2024.
>
>
> - **On Connection to Maximum Likelihood Training: ”Do you have any theoretical guarantees that minimizing the proximal matching loss will also improve—or even converge to—the true data log‐likelihood?”**
>
> **Response:** Thank you for this insightful question.
>
> 1) Since our model is mainly built upon the continuous-time SDE framework in [1], it does not naturally come with an MLE interpretation as originated from the discrete framework in DDPM [2]. It remains an open question whether such an equivalence exists between proximal matching and MLE training, and establishing such an equivalence can be non-trivial. For instance, in the derivation in [2], the KL term between Gaussian distributions reduces to an MSE between means—this is critical for linking denoising score matching to ELBO. In contrast, our proximal matching loss is not an MSE, and the transition kernel in ProxDM is non-Gaussian. Nonetheless, we believe this is a promising direction and plan to explore it in future work.
>
> 2) Although our method does not have a likelihood-based guarantee, it is supported by rigorous theoretical guarantees from a sample distribution perspective. In Theorem 1, we showed that if true proximal operators are known, the sample distribution generated by our algorithm converges to the true data distribution in KL divergence. Furthermore, prior work [3] establishes theoretical guarantees for learning true proximal operators via proximal matching. Together, these results provide strong theoretical support for our approach.
>
> 3) Additionally, as Gribonval [4] showed, MAP and MMSE denoisers can be closely related under a change of distributions. This suggests that our training paradigm may correspond to MLE training for a related distribution, which may be worth investigating in the future.
>
>
> References:
>
>
> [1] Song, et al. "Score-based generative modeling through stochastic differential equations." ICLR 2021.
>
> [2] Ho, et al. "Denoising diffusion probabilistic models." NeurIPS 2020.
>
> [3] Fang, et al. "What's in a prior? learned proximal networks for inverse problems." ICLR 2024.
>
> [4] Gribonval R. “Should Penalized Least Squares Regression be Interpreted as Maximum A Posteriori Estimation?” IEEE Trans. Signal Process. 2011.
>
>
>
> - **On Training Efficiency:**
>   - **”The paper reports faster convergence, but is that improvement limited to the sampling phase, or does it extend to faster convergence during training as well?”**
>   - **”Comparative analyses of training efficiency and convergence behavior are absent.”**
>
> **Response:** We will be happy to provide training curves (e.g., FID vs. steps) in the supplementary material. Based on our experiments, we don't observe a clear advantage in training convergence speed compared to standard diffusion models (neither do we have reasons to believe this should be the case, unlike in sampling). This can be because proximal diffusion models are conditioned on two parameters - time and lambda, whereas score models are conditioned on time only. This higher dimension requires proximal diffusion models to learn more complex functions. In the future, better training strategies may help accelerate training, such as better loss weighting or sampling for time and lambda, as commented in Section 3.3.

---

> > ### Comment · Reviewer_uiu8 · 2025-08-08
> > **Response**
> >
> > I appreciate the authors’ thorough response and additional experiments. Theoretically, I remain unconvinced that training a MAP denoiser offers an inherent advantage over directly training a denoiser on a likelihood bound for the data distribution. My current view is that the main effect is a tradeoff: faster sampling with more susceptibility to mode seeking and reduced sample diversity, analogous to DDIM versus DDPM sampling. Practically, I can imagine benefits, since improved density estimation in diffusion models does not reliably translate into better perceptual metrics.
> >
> > It is still unclear how the Proximal Diffusion model fits within the present diffusion and flow matching landscape, given the immense advancement in both theory and engineering. That said, I agree with the other reviewers that the paper is carefully derived, well written, and offers strong insights. In particular, framing the sampling procedure of diffusion models as an optimization problem, akin to SGD, was an engaging and thought-provoking perspective. Therefore, I will raise my score.

---

> ### Comment · Area_Chair_RMhK · 2025-08-05
>
> Dear reviewer,
> If not already, please take a look at the authors' rebuttal, and discuss if necessary.
> Thanks,
> -AC

---

> ### Author Response · Authors · 2025-08-07
>
> Dear reviewer,
>
> Thank you again for your constructive comments. Please kindly let us know if there are any remaining concerns or if our answers have addressed all your questions.
>
> Thank you!

---

### Decision · Program_Chairs · 2025-09-17

**Decision:**

Accept (poster)

**Comment:**

The inference of diffusion model is typically enabled by a time discretization of its backward process (such as a backward SDE), and this paper considers discretizing the backward SDE using backward Euler numerical scheme instead of the commonly used explicit schemes. This is equivalent to a proximal map formulation. However, backward Euler, being an implicit scheme, is computationally expensive, and the key idea of this paper is to learn the proximal map instead of the score function, so that the necessity of implicit equation solves needed by backward Euler can be circumvented. The efficacy of the proposed method is then demonstrated both theoretically and empirically. Both reviewers and I were impressed and convinced by the clarity and the extent of demonstration. While some reviewers also pointed out (and I agree) that a comparison to more recent theoretical results, such as [1-3], should be discussed, all of us agree that this work has enough merits for an acceptance, which I'm delighted to recommend. I encourage the authors to take the discussions into consideration when preparing for a revised/final version.

[1] Nearly d-linear convergence bounds for diffusion models via stochastic localization
[2] KL convergence guarantees for score diffusion models under minimal data assumptions
[3] Evaluating the design space of diffusion-based generative models
(note these are not based on advanced discretization, unlike the compared `accelerated strategy')